# Channels’ Confirmation and Predictions’ Confirmation: From the Medical Test to the Raven Paradox

**DOI:** 10.3390/e22040384

**Published:** 2020-03-26

**Authors:** Chenguang Lu

**Affiliations:** Intelligence Engineering and Mathematics Institute, Liaoning Technical University, Fuxin 123000, China; survival99@gmail.com

**Keywords:** relative entropy, cross-entropy, uncertain reasoning, inductive logic, confirmation measure, semantic information, medical test, raven paradox

## Abstract

After long arguments between positivism and falsificationism, the verification of universal hypotheses was replaced with the confirmation of uncertain major premises. Unfortunately, Hemple proposed the Raven Paradox. Then, Carnap used the increment of logical probability as the confirmation measure. So far, many confirmation measures have been proposed. Measure *F* proposed by Kemeny and Oppenheim among them possesses symmetries and asymmetries proposed by Elles and Fitelson, monotonicity proposed by Greco et al., and normalizing property suggested by many researchers. Based on the semantic information theory, a measure *b** similar to *F* is derived from the medical test. Like the likelihood ratio, measures *b** and *F* can only indicate the quality of channels or the testing means instead of the quality of probability predictions. Furthermore, it is still not easy to use *b**, *F*, or another measure to clarify the Raven Paradox. For this reason, measure *c** similar to the correct rate is derived. Measure *c** supports the Nicod Criterion and undermines the Equivalence Condition, and hence, can be used to eliminate the Raven Paradox. An example indicates that measures *F* and *b** are helpful for diagnosing the infection of Novel Coronavirus, whereas most popular confirmation measures are not. Another example reveals that all popular confirmation measures cannot be used to explain that a black raven can confirm “Ravens are black” more strongly than a piece of chalk. Measures *F*, *b**, and *c** indicate that the existence of fewer counterexamples is more important than more positive examples’ existence, and hence, are compatible with Popper’s falsification thought.

## 1. Introduction

A universal judgment is equivalent to a hypothetical judgment or a rule, such as “All ravens are black” is equivalent to “For every *x*, if *x* is a raven, then *x* is black”. Both can be used as a major premise for a syllogism. Deductive logic needs major premises; however, some major premises for empirical reasoning must be supported by inductive logic. Logical empiricism affirmed that a universal judgment can be verified finally by sense data. Popper said against logical empiricism that a universal judgment could only be falsified rather than be verified. However, for a universal or hypothetical judgment that is not strict, and is therefore uncertain, such as “Almost all ravens are black”, “Ravens are black”, or “If a man’s Coronavirus test is positive, then he is very possibly infected”, we cannot say that one counterexample can falsify it. After long arguments, Popper and most logical empiricists reached the identical conclusion [1,2] that we may use evidence to confirm universal judgments or major premises that are not strict or uncertain. 

In 1945, Hemple [3] proposed the confirmation paradox or the Raven Paradox. According to the Equivalence Condition in the classical logic, “If *x* is a raven, then *x* is black” (Rule I) is equivalent to “If *x* is not black, then *x* is not a raven” (Rule II). A piece of white chalk supports the Rule II, and hence, also supports the Rule I. However, according to the Nicod criterion [4], a black raven supports the Rule I, a non-black raven undermines the Rule I, and a non-raven thing, such as a black cat or a piece of white chalk, is irrelevant to the Rule I. Hence, there exists a paradox between the Equivalence Condition and the Nicod criterion. 

To quantize confirmation, both Carnap [1] and Popper [2] proposed their confirmation measures. However, only Carnap’s confirmation measures are famous. So far, researchers have proposed many confirmation measures [1,5,6,7,8,9,10,11,12,13]. The induction problem seemly has become the confirmation problem. To screen reasonable confirmation measures, Elles and Fitelson [14] proposed **symmetries and asymmetries** as desirable properties; Crupi et al. [8] and Greco et al. [15] suggested **normalization** (for measures between −1 and 1) as a desirable property; Greco et al. [16] proposed **monotonicity** as a desirable property. We can find that only measures *F* (proposed by Kemeny and Oppenheim) and *Z* among popular confirmation measures possess these desirable properties. Measure *Z* was proposed by Crupi et al. [8] as the normalization of some other confirmation measures. It is also called the certainty factor proposed by Shortliffe and Buchanan [7].

When the author of this paper researched semantic information theory [17], he found that an uncertain prediction could be treated as the combination of a clear prediction and a tautology; the combining proportion of the clear prediction could be used as the degree of belief; the degree of belief optimized with a sampling distribution could be regarded as a confirmation measure. This measure is denoted by *b**; it is similar to measure *F* and also possesses the above-mentioned desirable properties.

Good confirmation measures should possess not only mathematically desirable properties but also practicabilities. We can use medical tests to check their practicabilities. We use the degree of belief to represent the degree to which we believe a major premise and use the degree of confirmation to denote the degree of belief that is optimized by a sample or some examples. The former is subjective, whereas the latter is objective. A medical test provides the test-positive (or the test-negative) to predict if a person or a specimen is infected (or uninfected). Both the test-positive and the test-negative have degrees of belief and degrees of confirmation. In medical practices, there exists an important issue: if two tests provide different results, which test should we believe? For example, when both Nucleic Acid Test (NAT) and CT (Computed Tomography) are used to diagnose the infection of Novel Coronavirus Disease (COVID-19), if the result of NAT is negative and the result of CT is positive, which should we believe? According to the sensitivity and the specificity [18] of a test and the prior probability of the infection, we can use any confirmation measure to calculate the degrees of confirmation of the test-positive and the test-negative. Using popular confirmation measures, can we provide reasonable degrees of confirmation to help us choose a better result from NAT-negative and CT-positive? Can these degrees of confirmation reflect the probability of the infection?

This paper will show that only measures that are the functions of the likelihood ratio, such as *F* and *b**, can help us to diagnose the infection or choose a better result that can be accepted by the medical society. However, measures *F* and *b** do not reflect the probability of the infection. Furthermore, using *F*, *b**, or another measure, it is still difficult to eliminate the Raven Paradox. 

Recently, the author found that the problem with the Raven Paradox is different from the problem with the medical diagnosis. Measures *F* and *b** indicate how good the testing means are instead of how good the probability predictions are. To clarify the Raven Paradox, we need a confirmation measure that can indicate how good a probability prediction is. The confirmation measure *c** is hence derived. We call *c** a prediction confirmation measure and call *b** a channel confirmation measure. The distinction between Channels’ confirmation and predictions’ confirmation is similar to yet different from the distinction between Bayesian confirmation and Likelihoodist confirmation [19]. Measure *c** accords with the Nicod criterion and undermines the Equivalence Condition, and hence can be used to eliminate the Raven Paradox. 

The main purposes of this paper are:to distinguish channel confirmation measures that are compatible with the likelihood ratio and prediction confirmation measures that can be used to assess probability predictions,to use a prediction confirmation measure *c** to eliminate the Raven Paradox, andto explain that confirmation and falsification may be compatible.

The confirmation methods in this paper are different from popular methods, since:Measures *b** and *c** are derived by the semantic information method [17,20] and the maximum likelihood criterion rather than defined directly.Confirmation and statistical learning mutually support so that the confirmation measures can be used not only to assess major premises but also to make probability predictions.

The main contributions of this paper are:It clarifies that we cannot use one confirmation measure for two different tasks: (1) to assess (communication) channels, such as medical tests as testing means, and (2) to assess probability predictions, such as to assess “Ravens are black”.It provides measure *c** that manifests the Nicod criterion and hence provides a new method to clarify the Raven Paradox.

The rest of this paper is organized as follows. Section 2 includes background knowledge. It reviews existing confirmation measures, introduces the related semantic information method, and clarifies some questions about confirmation. Section 3 derives new confirmation measures *b** and *c** with the medical test as an example. It also provides many confirmation formulas for major premises with different antecedents and consequents. Section 4 includes results. It gives some cases to show the characteristics of new confirmation measures, to compare various confirmation measures by applying them to the diagnosis of COVID-19, and to show how an increased example affects the degrees of confirmation with different confirmation measures. Section 5 discusses why we can only eliminate the Raven Paradox by measure *c**. It also discusses some conceptual confusion and explains how new confirmation measures are compatible with Popper’s falsification thought. Section 5 ends with conclusions.

## 2. Background

### 2.1. Statistical Probability, Logical Probability, Shannon’s Channel, and Semantic Channel

First we distinguish logical probability and statistical probability. Logical probability of a hypothesis (or a label) is the probability in which the hypothesis is judged to be true, whereas its statistical probability is the probability in which the hypothesis or the label is selected. 

Suppose that ten thousand people go through a door. For everyone denoted by *x*, entrance guards judge if *x* is elderly. If two thousand people are judged to be elderly, then the logical probability of the predicate “*x* is elderly” is 2000/10,000 = 0.2. If the task of entrance guards is to select a label for every person from four labels: “Child”, “Youth”, “Adult”, and “Elderly”, there may be one thousand people who are labeled “Elderly”. The statistical probability of “Elderly” should be 1000/10,000 = 0.1. Why are not two thousand people are labeled “Elderly”? The reason is that some elderly people are labeled “Adult”. A person may make two labels be true, such as a 65 years old person makes both “Adult” and “Elderly” be true. That is why the logical probability of a label is often greater than its statistical probability. An extreme example is that the logical probability of a tautology, such as “*x* is elderly or not elderly”, is 1, whereas its statistical probability is almost 0 in general because a tautology is rarely selected. Statistical probability is normalized (the sum is 1), whereas logical probability is not normalized in general [17]. Therefore, we use two different symbols “*P*” and “*T*” to distinguish statistical probability and logical probability. 

We now consider the Shannon channel [21] between human ages and labels “Child”, “Adult”, “Youth”, “Middle age”, “Elderly”, and the like.

Let *X* be a random variable to denote an age and *Y* be a random variable to denote a label. *X* takes a value *x*∈{ages}; Y takes a value *y*∈{“Child”, “Adult”, “Youth”, “Middle age”, “Elderly”,…}. Shannon calls the prior probability distribution *P*(*X*) (or *P*(*x*)) the source, and calls *P*(*Y*) the destination. There is a Shannon channel *P*(*Y|X*) from *X* to *Y*. It is a transition probability matrix: (1)$$P(Y|X)\Leftrightarrow \left[\begin{array}{cccc} P(y_{1}|x_{1}) & P(y_{1}|x_{2}) & \ldots & P(y_{1}|x_{m})\\ P(y_{2}|x_{1}) & P(y_{2}|x_{2}) & \ldots & P(y_{2}|x_{m})\\ \ldots & \ldots & \ldots & \ldots \\ P(y_{n}|x_{1}) & P(y_{n}|x_{2}) & \ldots & P(y_{n}|x_{m}) \end{array}\right]\Leftrightarrow \left[\begin{array}{c} P(y_{j}|x)\\ P(y_{j}|x)\\ \ldots \\ P(y_{n}|x) \end{array}\right],$$
where ⇔ indicates equivalence. This matrix consists of a group of conditional probabilities *P*(*y_j_|x_i_*) (*j* = 0, 1, …, *n*; *i* = 0, 1, …, *m*) or a group of transition probability functions (so called by Shannon [21]), *P*(*y_j_|x*) (*j* = 0, 1, …, *n*), where *y_j_* is a constant, and *x* is a variable. 

There is also a semantic channel that consists of a group of truth functions. Let *T*(*θ_j_*|*x*) be the truth function of *y_j_*, where *θ_j_* is a model or a set of model parameters, by which we construct *T*(*θ_j_*|*x*). The *θ_j_* is alse explained as a fuzzy sub-set of the domain of *x* [17]. For example, *y_j_* = “*x* is young”. Its truth function may be
*T*(*θ_j_*|*x*) = exp[−(*x* − 20)^2^/25],(2)
where 20 and 25 are model parameters. For *y_k_* = “*x* is elderly”, its truth function may be a logistic function:*T*(*θ_k_*|*x*) = 1/[1 + exp[−0.2(*x* − 65)],(3)
where 0.2 and 65 are model parameters. The two truth functions are shown in Figure 1.

According to Tarski’s truth theory [22] and Davidson’s truth-conditional semantics [23], a truth function can represent the semantic meaning of a hypothesis. Therefore, we call the matrix, which consists of a group of truth functions, a semantic channel:(4)$$T(\theta |X)\Leftrightarrow \left[\begin{array}{cccc} T(\theta_{1}|x_{1}) & T(\theta_{1}|x_{2}) & \ldots & T(\theta_{1}|x_{m})\\ T(\theta_{2}|x_{1}) & T(\theta_{2}|x_{2}) & \ldots & T(\theta_{2}|x_{m})\\ \ldots & \ldots & \ldots & \ldots \\ T(\theta_{n}|x_{1}) & T(\theta_{n}|x_{2}) & \ldots & T(\theta_{n}|x_{m}) \end{array}\right]\Leftrightarrow \left[\begin{array}{c} T(\theta_{1}|x)\\ T(\theta_{2}|x)\\ \ldots \\ T(\theta_{n}|x) \end{array}\right].$$

Using a transition probability function *P*(*y_j_*|*x*), we can make the probability prediction *P*(*x*|*y_j_*) by
*P*(*x*|*y_j_*) = *P*(*x*)*P*(*y_j_*|*x*)/*P*(*y_j_*),(5)
which is the classical Bayes’ formula. Using a truth function *T*(*θ_j_*|*x*), we can also make a probability prediction or produce a likelihood function by
(6)$$P(x|\theta_{j})=P(x)T(\theta_{j}|x)/T(\theta_{j}),$$
where *T*(*θ_j_*) is the logical probability of *y_j_*. There is
(7)$$T(\theta_{j})=\sum_{i}P(x_{i})T(\theta_{j}|x_{i}).$$

Equation (6) is called the semantic Bayes’ formula [17]. The likelihood function is subjective; it may be regarded as the hybird of logical probability and statistical probability. 

When the source *P*(*x*) is changed, the above formulas for predictions still work. It is easy to prove that *P*(*x|θ_j_*) = *P*(*x|y_j_*) as *T*(*θ_j_*|*x*)∝*P*(*y_j_|x*). Since the maximum of *T*(*θ_j_*|*x*) is 1, letting *P*(*x|θ_j_*) = *P*(*x*|*y_j_*), we can obtain the optimized truth function [17]:*T**(*θ_j_*|*x*) = [*P*(*x*|*y_j_*)/*P*(*x*)]/max[*P*(*x*|*y_j_*)/*P*(*x*)] = *P*(*y_j_*|*x*)/max[*P*(*y_j_*|*x*)], (8)
where *x* is a variable and max(.) is the maximum of the function in brackets (.).

### 2.2. To Review Popular Confirmation Measures

We use *h*_1_ to denote a hypothesis, *h*_0_ to denote its negation, and *h* to denote one of them. We use *e*_1_ as another hypothesis as the evidence of *h*_1_, *e*_0_ as its negation, and *e* as one of them. We use *c*(*e*, *h*) to represent a confirmation measure, which means the degree of inductive support. Note that *c*(*e, h*) here is used as in [8], where *e* is on the left, and *h* is on the right. 

In the existing studies of confirmation, logical probability and statistical probability are not definitely distinguished. We still use *P* for both in introducing popular confirmation measures. 

The popular confirmation measures include:*D*(*e*_1_, *h*_1_)=*P*(*h*_1_|*e*_1_)−*P*(*h*_1_)     (Carnap, 1962 [1]),*M*(*e*_1_, *h*_1_) = *P*(*e*_1_|*h*_1_)−*P*(*e*_1_)     (Mortimer, 1988 [5]),*R*(*e*_1_, *h*_1_) = log[*P*(*h*_1_|*e*_1_)/*P*(*h*_1_)]     (Horwich, 1982 [6]),*C*(*e*_1_, *h*_1_) = *P*(*h*_1_, *e*_1_)−*P*(*e*_1_)*P*(*h*_1_)    (Carnap,1962 [1]),$Z(h_{1},e_{1})=\{\begin{array}{l} {}[P(h_{1}|e_{1})-P(h_{1})]/P(h_{0}),\;\mathrm{as}\;P(h_{1}|e_{1})\geq P(h_{1}),\\ {}[P(h_{1}|e_{1})-P(h_{1})]/P(h_{1}),\;\mathrm{otherwise}, \end{array}$(Shortliffe and Buchanan, 1975 [7], Crupi et al., 2007 [8]),S(*e*_1_, *h*_1_) = *P*(*h*_1_|*e*_1_)−*P*(*h*_1_|*e*_0_)              (Christensen, 1999 [9]),*N*(*e*_1_, *h*_1_) = *P*(*e*_1_|*h*_1_)−*P*(*e*_1_|*h*_0_)             (Nozik, 1981 [10]),*L*(*e*_1_, *h*_1_) = log[*P*(*e*_1_|*h*_1_)/*P*(*e*_1_|*h*_0_)]            (Good, 1984 [11]), and*F*(*e*_1_, *h*_1_) = [ *P*(*e*_1_|*h*_1_)−*P*(*e*_1_|*h*_0_)]/[ *P*(*e*_1_|*h*_1_)+ *P*(*e*_1_|*h*_0_)]    (Kemeny and Oppenheim, 1952 [12]).

Two measures *D* and *C* proposed by Carnap are for incremental confirmation and absolute confirmation respectively. There are more confirmation measures in [8,24]. Measure *F* is also denoted by *l** [13], *L* [8], or *k* [24]. Most authors explain that probabilities they use, such as *P*(*h*_1_) and *P*(*h*_1_|*e*_1_) in *D*, *R*, and *C*, are logical probabilities. Some authors explain that probabilities they use, such as *P*(*e*_1_|*h*_1_) in *F*, are statistical probabilities.

Firstly, we need to clarify that confirmation is to assess what kind of evidence supports what kind of hypotheses. Let us have a look at the following three hypotheses: Hypothesis 1: *h*_1_(*x*) = “*x* is elderly”, where *x* is a variable for an age and *h*_1_(*x*) is a predicate. An instance *x*=70 may be the evidence, and the truth value *T*(*θ*_1_|70) of proposition *h*_1_(70) should be 1. If *x*=50, the (uncertain) truth value should be less, such as 0.5. Let *e*_1_ = “*x* ≥ 60”, true *e*_1_ may also be the evidence that supports *h*_1_ so that *T*(*θ*_1_|*e*_1_) > *T*(*θ*_1_).Hypothesis 2: *h*_1_(*x*) = “If age *x* ≥ 60, then *x* is elderly”, which is a hypothetical judgment, a major premise, or a rule. Note that *x* = 70 or *x* ≥ 60 is only the evidence of the consequent “*x* is elderly” instead of the evidence of the rule. The rule’s evidence should be a sample with many examples.Hypothesis 3: *e*_1_→*h*_1_ = “If age *x* ≥ 60, then *x* is elderly”, which is the same as Hypothesis 2. The difference is that *e*_1_ = “*x* ≥ 60”; *h*_1_ = “*x* is elderly”. The evidence is a sample with many examples like {(*e*_1_, *h*_1_), (*e*_1_, *h*_0_), …}, or a sampling distribution *P*(*e, h*), where *P* means statistical probability.

Hypothesis 1 has a (uncertain) truth function or a conditional logic probability function between 0 and 1, which is ascertained by our definition or usage. Hypothesis 1 need not be confirmed. Hypothesis 2 or Hypothesis 3 is what we need to confirm. The degree of confirmation is between −1 and 1.

There exist two different understandings about *c*(*e*, *h*):Understanding 1: The *h* is the major premise to be confirmed, and *e* is the evidence that supports *h*; *h* and *e* are so used by Elles and Fitelson [14].Understanding 2: The *e* and *h* are those in rule *e*→*h* as used by Kemeny and Oppenheim [12]. The *e* is only the evidence that supports consequent *h* instead of the major premise *e*→*h* (see Section 2.3 for further analysis).

Fortunately, although researchers understand *c*(*e*, *h*) in different ways, most researchers agree to use a sample including four types of examples (*e*_1_, *h*_1_), (*e*_0_, *h*_1_), (*e*_1_, *h*_0_), and (*e*_0_, *h*_0_) as the evidence to confirm a rule and to use the four examples’ numbers *a, b, c*, and *d* (see Table 1) to construct confirmation measures. The following statements are based on this common view.

The *a* is the number of example (*e*_1_, *h*_1_). For example, *e*_1_ = “raven” (“raven” is a label or the abbreviate of “*x* is a raven”) and *h*_1_ = “black”; *a* is the number of black ravens. Similarly, *b* is the number of black non-raven things; *c* is the number of non-black ravens; *d* is the number of non-black and non-raven things.

To make the confirmation task clearer, we follow Understanding 2 to treat *e*→*h* = “if *e* then *h*” as the rule to be confirmed and replace *c*(*e*, *h*) with *c*(*e*→*h*). To research confirmation is to construct or select the function *c*(*e*→*h*)=*f*(*a*, *b*, *c*, *d*). 

To screen reasonable confirmation measures, Elles and Fitelson [14] propose the following symmetries:Hypothesis Symmetry (HS): *c*(*e*_1_→*h*_1_) = −*c*(*e*_1_→*h*_0_) (two consequents are opposite),Evidence Symmetry (ES): *c*(*e*_1_→*h*_1_) = −*c*(*e*_0_→*h*_1_) (two antecedents are opposite),Commutativity Symmetry (CS): *c*(*e*_1_→*h*_1_) = *c*(*h*_1_→*e*_1_), andTotal Symmetry (TS): *c*(*e*_1_→*h*_1_) = *c*(*e*_0_→*h*_0_).

They conclude that only HS is desirable; the other three symmetries are not desirable. We call this conclusion the symmetry/asymmetry requirement. Their conclusion is supported by most researchers. Since TS is the combination of HS and ES, we only need to check HS, ES, and CS. According to this symmetry/asymmetry requirement, only measures *L*, *F*, and *Z* among the measures mentioned above are screened out. It is uncertain whether *N* can be ruled out by this requirement [15]. See [14,25,26] for more discussions about the symmetry/asymmetry requirement. 

Greco et al. [15] propose monotonicity as a desirable property. If *f*(*a*, *b*, *c*, *d*) does not decrease with *a* or *d* and does not increase with *b* or *c*, then we say that *f*(*a*, *b*, *c*, *d*) has the monotonicity. Measures *L*, *F*, and *Z* have this monotonicity, whereas measures *D*, *M*, and *N* do not have. If we further require that *c*(*e→h*) are normalizing (between −1 and 1) [8,12], then only *F* and *Z* are screened out. There are also other properties discussed [15,19]. One is logicality, which means *c*(*e→h*) = 1 without counterexample and *c*(*e→h*) = −1 without positive example. We can also screen out *F* and *Z* using the logicality requirement. 

Consider the medical test, such as the test for COVID-19. Let *e*_1_ = “positive” (e.g., “*x* is positive”, where *x* is a specimen), *e*_0_ = “negative”, *h*_1_ = “infected” (e.g.,“*x* is infected”), and *h*_0_ = “uninfected”. Then the positive likelihood ratio is *LR*^+^ = *P*(*e*_1_|*h*_1_)/*P*(*e*_1_|*h*_0_), which indicates the reliability of the rule *e*_1_→*h*_1_. Measures *L* and *F* have the one-to-one correspondence with *LR*:*L*(*e*_1_→*h*_1_) = log *LR*^+^;(9)
*F*(*e*_1_, *h*_1_)=(*LR*^+^ − 1)/(*LR*^+^ + 1).(10)

Hence, *L* and *F* can also be used to assess the reliability of the medical test. In comparison with *LR* and *L*, *F* can indicate the distance between a test (any *F*) and the best test (*F* = 1) or the worst test (*F* = −1) better than *LR* and *L*. However, *LR* can be used for the probability predictions of diseases more conveniently [27].

### 2.3. To Distinguish a Major Premise’s Evidence and Its Consequent’s Evidence

The evidence for the consequent of a syllogism is the minor premise, whereas the evidence for a major premise or a rule is a sample or a sampling distribution *P*(*e*, *h*). In some researchers’ studies, *e* is used sometimes as the minor premise, and sometimes as an example or a sample; *h* is used sometimes as a consequent, and sometimes as a major premise. Researchers use *c*(*e*, *h*) or *c*(*h*, *e*) instead of *c*(*e*→*h*) because they need to avoid the contradiction between the two understandings. However, if we distinguish the two types of evidence, it has no problem to use *c*(*e*→*h*). We only need to emphasize that the evidence for a major premise is a sampling distribution *P*(*e*, *h*) instead of *e*.

If *h* is used as a major premise and *e* is used as the evidence (such as in [14,28]), −*e* (the negation of *e*) is puzzling because there are four types of examples instead of two. Suppose *h* = *p*→*q* and that *e* is one of (*p*, *q*), (*p*, −*q*), (−*p*, *q*), and (−*p*, *q*). If (*p*, −*q*) is the counterexample, and other three examples (*p*, *q*), (−*p*, *q*) and (−*p*, −*q*) are positive examples, which support *p*→*q*, then (−*p*, *q*) and (−*p*, −*q*) should also support *p*→−*q* because of the same reason. However, according to HS [14], it is unreasonable that the same evidence supports both *p*→*q* and *p*→−*q*. In addition, *e* is a sample with many examples in general. A sample’s negation or a sample’s probability is also puzzling.

Fortunately, though many researchers say that *e* is the evidence of a major premise *h*, they also treat *e* as the antecedent and treat *h* as the consequent of a major premise because, only in this way, one can calculate the probabilities or conditional probabilities of *e* and *h* for a confirmation measure. Why, then, should we replace *c*(*e*, *h*) with *c*(*e*→*h*) to make the task clearer? Section 5.3 will show that *h* used as a major premise will result in the misunderstanding of the symmetry/asymmetry requirement.

### 2.4. Incremental Confirmation or Absolute Confirmation

Confirmation is often explained as assessing the impact of evidence on hypotheses, or the impact of the premise on the consequent of a rule [14,19]. However, this paper has a different point of view that confirmation is to assess how well a sample or sampling distribution supports a major premise or a rule; the impact on the rule (e.g., the increment of degree of confirmation) may be made by newly added examples.

Since one can use one or several examples to calculate the degree of confirmation with a confirmation measure, many researchers call their confirmation incremental confirmation [14,15]. There are also researchers who claim that we need absolute confirmation [29]. This paper supports absolute confirmation.

The problem with incremental confirmation is that the degrees of confirmation calculated are often bigger than 0.5 and are irrelevant to our prior knowledge or *a, b, c*, and *d* that we knew before. It is unreasonable to ignore prior knowledge. Suppose that the logical probability of *h*_1_ = “*x* is elderly” is 0.2; the evidence is one or several people with age(s) *x* > 60; the conditionally logical probability of *h*_1_ is 0.9. With measure *D*, the degree of confirmation is 0.9 − 0.2 = 0.7, which is very large and irrelevant to the prior knowledge.

In confirmation function *f*(*a*, *b*, *c*, *d*), the numbers *a*, *b*, *c*, and *d* should be those of all examples including past and current examples. A measure *f*(*a*, *b*, *c*, *d*) should be an absolute confirmation measure. Its increment should be
Δ*f* = *f*(*a* + Δ*a*, *b* + Δ*b*, *c* +Δ*c*, *d* + Δ*d*) − *f*(*a*, *b*, *c*, *d*).(11)

The increment of the degree of confirmation brought about by a new example is closely related to the number of old examples. Section 5.2 will further discuss incremental confirmation and absolute confirmation.

### 2.5. The Semantic Channel and the Degree of Belief of Medical Tests

We now consider the Shannon channel and the semantic channel of the medical test. The relation between *h* and *e* is shown in Figure 2.

In Figure 2, *h*_1_ denotes an infected specimen (or person), *h*_0_ denotes an uninfected specimen, *e*_1_ is positive, and *e*_0_ is negative. We can treat *e*_1_ as a prediction “*h* is infected” and *e*_0_ as a prediction “*h* is uninfected”. In other word, *h* is a true label or true statement, and *e* is a prediction or selected label. The *x* is the observed feature of *h*; *E*_1_ and *E*_2_ are two sub-sets of the domain of *x*. If *x* is in *E*_1_, then *e*_1_ is selected; if *x* is in *E*_0_, then *e*_0_ is selected. 

Figure 3 shows the relationship between *h* and *x* by two posterior probability distributions *P*(*x*|*h*_0_) and *P*(*x*|*h*_1_) and the magnitudes of four conditional probabilities (with four colors). 

In the medical test, *P*(*e*_1_|*h*_1_) is called sensitivity [18], and *P*(*h*_0_|*e*_0_) is called specificity. They ascertain a Shannon channel, which is denoted by *P*(*e|h*), as shown in Table 2.

We regard predicate *e*_1_(*h*) as the combination of believable and unbelievable parts (see Figure 4). The truth function of the believable part is *T*(*E*_1_|*h*)∈{0,1}. The unbelievable part is a tautology, whose truth function is always 1. Then we have the truth functions of predicates *e*_1_(*h*) and *e*_0_(*h*):*T*(*θ_e_*_1_*|h*)= *b*_1_’ + *b*_1_*’ T*(*E*_1_*|h*); *T*(*θ_e_*_0_|*h*) = *b*_0_’ + *b*_0_*’ T*(*E*_0_|*h*).(12)
where model parameter *b*_1_’ is the proportion of the unbelievable part, and also the truth value for the counter-instance *h*_0_.

The four truth values form a semantic channel, as shown in Table 3.

For medical tests, the logical probability of *e*_1_ is
(13)$$T(\theta_{e1})=\sum_{i}P(h_{i})T(\theta_{e1}|h_{i})=P(h_{1})+b_{1}\prime P(h_{0}),$$

The likelihood function is
(14)$$P(h|\theta_{e1})=P(h)T(\theta_{e1}|h)/T(\theta_{e1}).$$*P*(*h|θ_j_*) is also the predicted probability of *h* according to *T*(*θ_e_*_1_|*h*) or the semantic meaning of *e*_1_.

To measure subjective or semantic information, we need subjective probability or logical probability [17]. To measure confirmation, we need statistical probability.

### 2.6. Semantic Information Formulas and the Nicod–Fisher Criterion

According to the semantic information G theory [17], the (amount of) semantic information conveyed by *y_j_* about *x_i_* is defined with the log-normalized-likelihood: (15)$$I(x_{i};\theta_{j})=\log\frac{P(x_{i}|\theta_{j})}{P(x_{i})}=\log\frac{T(\theta_{j}|x_{i})}{T(\theta_{j})},$$
where *T*(*θ_j_*|*x_i_*) is the truth value of proposition *y_j_*(*x_i_*) and *T*(*θ_j_*) is the logical probability of *y_j_*. If *T*(*θ_j_*|*x*) is always 1, then this semantic information formula becomes Carnap and Bar-Hillel’s semantic information formula [30].

In semantic communication, we often see hypotheses or predictions, such as “The temperature is about 10 ˚C”, “The time is about seven o’clock”, or “The stock index will go up about 10% next month”. Each one of them may be represented by *y_j_* = “*x* is about *x_j_*.” We can express the truth functions of *y_j_* by
*T*(*θ_j_*|*x*) = exp[−(*x* − *x_j_*)^2^/(2σ^2^)].(16)

Introducing Equation (16) into Equation (15), we have
(17)$$I(x_{i};\theta_{j})=\log[1/T(\theta_{j})]-{\left(x_{i}-x_{j}\right)}^{2}/(2\sigma^{2}),$$
by which we can explain that this semantic information is equal to the Carnap–Bar-Hillel’s semantic information minus the squared relative deviation. This formula is illustrated in Figure 5.

Figure 5 indicates that the smaller the logical probability is, the more information there is; and the larger the deviation is, the less information there is. Thus, a wrong hypothesis will convey negative information. These conclusions accord with Popper’s thought (see [2], p. 294). 

To average *I*(*x_i_*; *θ_j_*), we have generalized Kullback–Leibler information or relative cross-entropy:(18)$$I(X;\theta_{j})=\sum_{i}P(x_{i}|y_{j})\log\frac{P(x_{i}|\theta_{j})}{P(x_{i})}=\sum_{i}P(x_{i}|y_{j})\log\frac{T(\theta_{j}|x_{i})}{T(\theta_{j})},$$
where *P*(*x*|*y_j_*) is the sampling distribution, and *P*(*x*|*θ_j_*) is the likelihood function. If *P*(*x*|*θ_j_*) is equal to *P*(*x*|*y_j_*), then *I*(*X*; *θ_j_*) reaches its maximum and becomes the relative entropy or the Kullback–Leibler divergence.

Consider medical tests, the semantic information conveyed by *e*_1_ about *h* becomes
(19)$$I(h_{i};\theta_{e1})=\log\frac{P(h_{i}|\theta_{e1})}{P(h_{i})}=\log\frac{T(\theta_{e1}|h)}{T(\theta_{e1})}.$$

The average semantic information is:(20)$$I(h;\theta_{e1})=\sum_{i=0}^{1}P(h_{i}|e_{1})\log\frac{P(h_{i}|\theta_{e1})}{P(h_{i})}=\sum_{i=0}^{1}P(h_{i}|e_{1})\log\frac{T(\theta_{e1}|h_{i})}{T(\theta_{e1})}$$
where *P*(*h_i_|e*_1_) is the conditional probability from a sample. 

We now consider the relationship between the likelihood and the average semantic information.

Let **D** be a sample {(*h*(*t*), *e*(*t*))|*t* = 1 to *N*; *h*(*t*)∈{*h*_0_, *h*_1_}; *e*(*t*)∈{*e*_0_, *e*_1_}}, which includes two sub-samples or conditional samples **H**_0_ with label *e*_0_ and **H**_1_ with label *e*_1_. When *N* data points in **D** come from Independent and Identically Distributed random variables, we have the log-likelihood
(21)$$\begin{array}{c} L(\theta_{e1})=\log P(H_{1}|\theta_{e1})=\log P(h(1),h(2),\ldots ,h(N)|\theta_{e1})=\log\prod_{i=0}^{1}P{\left(h_{i}|\theta_{e1}\right)}^{N_{1i}}\\ =N_{1}\sum_{i=0}^{1}P(h_{i}|e_{1})\log P(h_{i}|\theta_{ej})=-N_{1}H(h|\theta_{e1}). \end{array}$$
where *N*_1*i*_ is the number of example (*h_i_, e*_1_) in **D**; *N*_1_ is the size of **H**_1_. *H*(*h|**θ_e_*_1_) is the cross-entropy. If *P*(*h*|*θ_e_*_1_) = *P*(*h*|*e*_1_), then the cross-entropy becomes the Shannon entropy. Meanwhile, the cross-entropy reaches its minimum, and the likelihood reaches its maximum. 

Comparing the above two equations, we have
(22)$$I(h;\theta_{e1})=L(\theta_{e1})/N_{1}-\sum_{i=0}^{1}P(h_{i}|e_{1})\log P(h_{i})$$
which indicates the relationship between the average semantic information and the likelihood. Since the second term on the right side is constant, the maximum likelihood criterion is equivalent to the maximum average semantic information criterion. It is easy to find that a positive example (*e*_1_, *h*_1_) increases the average log-likelihood *L*(*θ_e_*_1_)/*N*_1_; a counterexample (*e*_1_, *h*_0_) decreases it; examples (*e*_0_, *h*_0_) and (*e*_0_, *h*_1_) with *e*_0_ are irrelevant to it.

The Nicod criterion about confirmation is that a positive example (*e*_1_, *h*_1_) supports rule *e*_1_→*h*_1_; a counterexample (*e*_1_, *h*_0_) undermines *e*_1_→*h*_1_. No reference exactly indicates if Nicod affirmed that (*e*_0_, *h*_1_) and (*e*_0_, *h*_1_) are irrelevant to *e*_1_→*h*_1_. If Nicod did not affirm, we can add this affirmation to the criterion, then call the corresponding criterion the Nicod–Fisher criterion, since Fisher proposed the maximum likelihood estimation. From now on, we use the Nicod–Fisher criterion to replace the Nicod criterion.

### 2.7. Selecting Hypotheses and Confirming Rules: Two Tasks from the View of Statistical Learning

Researchers have noted the similarity between most confirmation measures and information measures. One explanation [31] is that information is the average of confirmatory impact. However, this paper gives a different explanation as follows. 

There are three tasks in statistical learning: label learning, classification, and reliability analysis. There are similar tasks in inductive reasoning:Induction. It is similar to label learning. For uncertain hypotheses, label learning is to train a likelihood function *P*(*x|θ_j_*) or a truth function *T*(*θ_j_|x*) by a sampling distribution [17]. The Logistic function often used for binary classifications may be treated as a truth function.Hypothesis selection. It is like classification according to different criteria.Confirmation. It is similar to reliability analysis. The classical methods are to provide likelihood ratios and correct rates (including false rates, as those in Table 8).

Classification and reliability analysis are two different tasks. Similarly, hypothesis selection and confirmation are two different tasks.

In statistical learning, classification depends on the criterion. The often-used criteria are the maximum posterior probability criterion (which is equivalent to the maximum correctness criterion) and the maximum likelihood criterion (which is equivalent to the maximum semantic information criterion [17]). The classifier for binary classifications is
(23)$$e(x)=\{\begin{array}{l} e_{1},\;\mathrm{if}\;P(\theta_{1}|x)\geq P(\theta_{0}|x),\;P(x|\theta_{1})\geq P(x|\theta_{0}),\;\mathrm{or}\;I(x;\theta_{1})\geq I(x;\theta_{0});\\ \;e_{0},\;\mathrm{otherwise}. \end{array}$$

After the above classification, we may use information criterion to assess how well *e_j_* is used to predict *h_j_*:(24)$$\begin{array}{l} I^{*}(h_{j};\theta_{ej})=I(h_{j};e_{j})=\log\frac{P(h_{j}|e_{j})}{P(h_{j})}=\log\frac{P(e_{j}|h_{j})}{P(e_{j})}\\ =\log P(h_{j}|e_{j})-\log P(h_{j})=\log P(e_{j}|h_{j})-\log P(e_{j})\\ =\log P(h_{j},e_{j})-\log[P(h_{j})P(e_{j})], \end{array}$$
where *I** means optimized semantic information. With information amounts *I*(*h_i_; θ_ej_*) (*i, j* = 0,1), we can optimize the classifier [17]: (25)$$e_{j}^{*}=f(x)=\underset{e_{j}}{\arg\max}[P(h_{0}|x)I(h_{0};\theta_{ej})+P(h_{1}|x)I(h_{1};\theta_{ej})].$$

The new classifier will provide the new Shannon’s channel *P*(*e|h*). The maximum mutual information classification can be achieved by repeating Equations (23) and (25) [17,32]. 

With the above classifiers, we can make prediction *e_j_* = “*x* is *h_j_*” according to *x*. To tell information receivers how reliable the rule *e_j_*→*h_j_* is, we need the likelihood ratio *LR* to indicate how good the channel is or need the correct rate to indicate how good the probability prediction is. Confirmation is similar. We need to provide a confirmation measure similar to *LR*, such as *F*, and a confirmation measure similar to the correct rate. The difference is that the confirmation measures should change between −1 and 1.

According to above analyses, it is easy to find that confirmation measures *D, N, R*, and *C* are more like information measures for assessing and selecting predictions instead of confirming rules. *Z* is their normalization [8]; it seems between an information measure and a confirmation measure. However, confirming rules is different from measuring predictions’ information; it needs the proportions of positive examples and counterexamples.

## 3. Two Novel Confirmation Measures

### 3.1. To Derive Channel Confirmation Measure b*

We use the maximum semantic information criterion, which is consistent with the maximum likelihood criterion, to derive the channel confirmation measure. According to Equations (13) and (18), the average semantic information conveyed by *e*_1_ about *h* is
(26)$$I(h;\theta_{e1})=P(h_{0}|e_{1})\log\frac{b_{1}^{\prime }}{P(h_{1}+b_{1}^{\prime }P(h_{0})}+P(h_{1}|e_{1})\log\frac{1}{P(h_{1}+b_{1}^{\prime }P(h_{0})}$$

Letting d*I*(*h;θ_e_*_1_)/d*b*_1_’ = 0, we can obtain the optimized *b*_1_’:(27)$$b_{1}^{\prime *}=\frac{P(h_{0}|e_{1})}{P(h_{0})}/\frac{P(h_{1}|e_{1})}{P(h_{1})},$$
where *P*(*h*_1_|*e*_1_)/ *P*(*h*_1_) ≥ *P*(*h*_0_|*e*_1_)/ *P*(*h*_0_). The *b*’* can be called a disconfirmation measure. Letting both the numerator and the denominator multiply by *P*(*e*_1_), the above formula becomes:*b*_1_’* = *P*(*e*_1_|*h*_0_)/ *P*(*e*_1_|*h*_1_) = (1 − specificity)/sensibility = 1/*LR*^+^.(28)

According to the semantic information G theory [17], when a truth function is proportional to the corresponding transition probability function, e.g., *T**(*θ_e_*_1_*|h*)∝*P*(*e*_1_|*h*), the average semantic information reaches its maximum. Using *T**(*θ_e_*_1_*|h*)∝*P*(*e*_1_|*h*), we can directly obtain
(29)$$\frac{b_{1}^{\prime *}}{P(e_{1}|h_{0})}=\frac{1}{P(e_{1}|h_{1})}$$
and Equation (28). We call
*b*_1_* = 1 − *b*_1_’* = [*P*(*e*_1_|*h*_1_) − *P*(*e*_1_|*h*_0_)]/*P*(*e*_1_|*h*_1_)(30)
the degree of confirmation of the rule *e*_1_→*h*_1_. Considering *P*(*e*_1_|*h*_1_) < *P*(*e*_1_|*h*_0_), we have
*b*_1_* = *b*_1_’* − 1 = [*P*(*e*_1_|*h*_0_) − *P*(*e*_1_|*h*_1_)]/*P*(*e*_1_|*h*_0_).(31)

Combining the above two formulas, we obtain
(32)$$b_{1}^{*}=b^{*}(e_{1}\to h_{1})=\frac{P(e_{1}|h_{1})-P(e_{1}|h_{0})}{\max[P(e_{1}|h_{1}),P(e_{1}|h_{0})]}=\frac{LR^{+}-1}{\max(LR^{+},1)}.$$

Since
(33)$$b_{1}^{*}=b^{*}(e_{1}\to h_{0})=\frac{P(e_{1}|h_{0})-P(e_{1}|h_{1})}{\max[P(e_{1}|h_{0}),P(e_{1}|h_{1})]}=-b^{*}(e_{1}\to h_{1}),$$
the *b*_1_* possesses HS or Consequent Symmetry.

In the same way, we obtain
(34)$$b_{0}^{*}=b^{*}(e_{0}\to h_{0})=\frac{P(e_{0}|h_{0})-P(e_{0}|h_{1})}{\max[P(e_{0}|h_{0}),P(e_{0}|h_{1})]}=\frac{LR^{-}-1}{\max(LR^{-},1)}.$$

Using Consequent Symmetry, we can obtain *b**(*e*_1_→*h*_0_) = −*b**(*e*_1_→*h*_1_) and *b**(*e*_0_→*h*_1_) = −*b**(*e*_0_→*h*_0_).

Using measure *b** or *F*, we can answer the question: if the result of NAT is negative and the result of CT is positive, which should we believe? Section 4.2 will provide the answer that is consistent with the improved diagnosis of COVID-19 in Wuhan.

Compared with *F*, *b** is better for probability predictions. For example, from *b*_1_* > 0 and *P*(*h*), we obtain
*P*(*h*_1_|*θ_e_*_1_) = *P*(*h*_1_)/[ *P*(*h*_1_) + *b*_1_’**P*(*h*_0_)] = *P*(*h*_1_)/[1 − *b*_1_**P*(*h*_0_)].(35)

This formula is much simpler than the classical Bayes’ formula (see Equation (5)).

If *b*_1_* = 0, then *P*(*h*_1_|*θ_e_*_1_) = *P*(*h*_1_). If *b*_1_* < 0, then we can make use of HS or Consequent Symmetry to obtain *b*_10_* = *b*_1_*(*e*_1_→*h*_0_) = |*b*_1_*(*e*_1_→*h*_1_)| = |*b*_1_*|. Then we have
*P*(*h*_0_|*θ_e_*_1_) = *P*(*h*_0_)/[ *P*(*h*_0_) + *b*_10_’**P*(*h*_1_)] = *P*(*h*_0_)/[1 − *b*_10_**P*(*h*_1_)].(36)

We can also obtain *b*_1_* = 2*F*_1_/(1 + *F*_1_) from *F*_1_ = *F*(*e*_1_→*h*_1_) for the probability prediction *P*(*h*_1_|*θ_e_*_1_), but the calculation of probability predictions with *F*_1_ is a little complicated.

So far, it is still problematic to use *b**, *F*, or another measure to handle the Raven Paradox. For example, as shown in Table 13, the increment of *F*(*e*_1_→*h*_1_) caused by Δ*d* = 1 is 0.348 − 0.333, whereas the increment caused by Δ*a* = 1 is 0.340 − 0.333. The former is greater than the latter, which means that a piece of white chalk can support “Ravens are black” better than a black raven. Hence measure *F* does not accord with the Nicod–Fisher criterion. Measures *b** and *Z* do not either.

Why does not measure *b** and *F* accord with the Nicod–Fisher criterion? The reason is that the likelihood *L*(*θ_e_*_1_) is related to prior probability *P*(*h*), whereas *b** and *F* are irrelevant to *P*(*h*).

### 3.2. To Derive Prediction Confirmation Measure c*

Statistics not only uses the likelihood ratio to indicate how reliable a testing means (as a channel) is but also uses the correct rate to indicate how reliable a probability prediction is. Measure *F* and *b** like *LR* cannot indicate the quality of a probability prediction. Most other measures have similar problems.

For example, we assume that an NAT for COVID-19 [33] has sensitivity *P*(*e*_1_|*h*_1_) = 0.5 and specificity *P*(*e*_0_|*h*_0_) = 0.95. We can calculate *b*_1_’* = 0.1 and *b*_1_* = 0.9. When the prior probability *P*(*h*_1_) of the infection changes, predicted probability *P*(*h*_1_|*θ_e_*_1_) (see Equation (35)) changes with the prior probability, as shown in Table 4. We can obtain the same results using the classical Bayes’ formula (see Equation (5)).

Data in Table 4 show that measure *b** cannot indicate the quality of probability predictions. Therefore, we need to use *P*(*h*) to construct a confirmation measure that can reflect the correct rate. 

We now treat probability prediction *P*(*h*|*θ_e_*_1_) as the combination of a believable part with proportion *c*_1_ and an unbelievable part with proportion *c*_1_’, as shown in Figure 6. We call *c*_1_ the degree of belief of the rule *e*_1_→*h*_1_ as a prediction. 

When the prediction accords with the fact, e.g., *P*(*h*|*θ_e_*_1_) = *P*(*h*|*e*_1_), *c*_1_ becomes *c*_1_*. The degree of disconfirmation for predictions is
*c*’*(*e*_1_→*h*_1_) = *P*(*h*_0_|*e*_1_)/*P*(*h*_1_|*e*_1_), if *P*(*h*_0_|*e*_1_) ≤ *P*(*h*_1_|*e*_1_);*c*’*(*e*_1_→*h*_1_) = *P*(*h*_1_|*e*_1_)/*P*(*h*_0_|*e*_1_), if *P*(*h*_1_|*e*_1_) ≤ *P*(*h*_0_|*e*_1_). (37)

Further, we have the prediction confirmation measure
(38)$$\begin{array}{l} c_{1}^{*}=c^{*}(e_{1}\to h_{1})=\frac{P(h_{1}|e_{1})-P(h_{0}|e_{1})}{\max(P(h_{1}|e_{1}),P(h_{0}|e_{1}))}\\ =\frac{2P(h_{1}|e_{1})-1}{\max(P(h_{1}|e_{1}),1-P(h_{1}|e_{1}))}=\frac{2CR_{1}-1}{\max(CR_{1},1-CR_{1})}. \end{array}$$
where *CR*_1_ = *P*(*h*_1_|*θ_e_*_1_) = *P*(*h*_1_|*e*_1_) is the correct rate of rule *e*_1_→*h*_1_. This correct rate means that the probability of *h*_1_ we predict as *x*∈*E*_1_ is *CR*_1_. Letting both the numerator and denominator of Equation (38) multiply by *P*(*e*_1_), we obtain
(39)$$c_{1}^{*}=c^{*}(e_{1}\to h_{1})=\frac{P(h_{1},e_{1})-P(h_{0},e_{1})}{\max(P(h_{1},e_{1}),P(h_{0},e_{1}))}=\frac{a-c}{\max(a,c)}.$$

The sizes of four areas covered by two curves in Figure 7 may represent *a*, *b*, *c*, and *d*. 

In like manner, we obtain
(40)$$c_{0}^{*}=c^{*}(e_{0}\to h_{0})=\frac{P(h_{0},e_{0})-P(h_{1},e_{0})}{\max(P(h_{0},e_{0}),P(h_{1},e_{0}))}=\frac{d-b}{\max(d,b)}.$$

Making use of Consequent Symmetry, we can obtain *c**(*e*_1_→*h*_0_) = −*c**(*e*_1_→*h*_1_) and *c**(*e*_0_→*h*_1_) = −*c**(*e*_0_→*h*_0_). 

In Figure 7, the sizes of the two areas covered by two curves are *P*(*h*_0_) and *P*(*h*_1_), which are different. If *P*(*h*_0_) = *P*(*h*_1_) = 0.5, then prediction confirmation measure *c** is equal to channel confirmation measure *b**.

Using measure *c**, we can directly assess the quality of the probability predictions. For *P*(*h*_1_|*θ_e_*_1_) = 0.77 in Table 4, we have *c*_1_* = (0.77 − 0.23)/0.77 = 0.701. We can also use *c** for probability predictions. When *c*_1_* > 0, according to Equation (39), we have the correct rate of rule *e*_1_→*h*_1_:(41)$$CR_{1}=P(h_{1}|\theta_{e1})=1/(1+c_{1}^{\prime *})=1/(2-c_{1}^{*})$$

For example, if *c*_1_* = 0.701, then *CR*_1_ = 1/(2−0.701) = 0.77. If *c**(*e*_1_→*h*_1_) = 0, then *CR*_1_ = 0.5. If *c**(*e*_1_→*h*_1_) < 0, we may make use of HS to have *c*_10_* = *c**(*e*_1_→*h*_0_) = |*c**_1_|, and then make probability prediction:(42)$$\begin{array}{l} P(h_{0}|\theta_{e1})=1/(2-c_{10}{}^{*}),\\ P(h_{1}|\theta_{e1})=1-P(h_{0}|\theta_{e1})=(1-c_{10}{}^{*})/(2-c_{10}{}^{*}). \end{array}$$

We may define another prediction confirmation measure by replacing operation max( ) with +:(43)$$\begin{array}{l} c_{F1}=c_{F}{}^{*}(e_{1}\to h_{1})=\frac{P(h_{1}|e_{1})-P(h_{0}|e_{1})}{P(h_{1}|e_{1})+P(h_{0}|e_{1})}=P(h_{1}|e_{1})-P(h_{0}|e_{1})\\ =\frac{P(h_{1},e_{1})-P(h_{0},e_{1})}{P(e_{1})}=\frac{a-c}{a+c}. \end{array}$$

The *c_F_** is also convenient for probability predictions when *P*(*h*) is certain. There is
(44)$$\begin{array}{l} P(h_{1}|\theta_{e1})=CR_{1}=(1+c_{F}{}_{1}{}^{*})/2;\\ P(h_{0}|\theta_{e1})=1-CR_{1}=(1-c_{F1}{}^{*})/2. \end{array}$$

However, when *P*(*h*) is variable, we should still use *b** with *P*(*h*) for probability predictions.

It is easy to prove that *c**(*e*_1_→*h*_1_) and *c_F_**(*e*_1_→*h*_1_) possess all the above-mentioned desirable properties.

### 3.3. Converse Channel/Prediction Confirmation Measures b*(h→e) and c*(h→e) 

Greco et al. [19] divide confirmation measures into
Bayesian confirmation measures with *P*(*h*|*e*) for *e→h*,Likelihoodist confirmation measures with *P*(*e|h*) for *e→h*,converse Bayesian confirmation measures with *P*(*h|e*) for *h→e*, andconverse Likelihoodist confirmation measures with *P*(*e|h*) for *h→e*.

Similarly, this paper divides confirmation measures into
channel confirmation measure *b**(*e→h*),prediction confirmation measure *c**(*e→h*),converse channel confirmation measure *b**(*h→e*), andconverse prediction confirmation measure *c**(*h→e*).

We now consider *c**(*h*_1_→*e*_1_). The positive examples’ proportion and the counterexamples’ proportion can be found in the upside of Figure 7. Then we have
(45)$$c^{*}(h_{1}\to e_{1})=\frac{P(e_{1}|h_{1})-P(e_{0}|h_{1})}{\max(P(e_{1}|h_{1}),P(e_{0}|h_{1}))}=\frac{a-b}{\max(a,b)}.$$

The correct rate reflected by *c**(*h*_1_→*e*_1_) is sensitivity or true positive rate *P*(*h*_1_|*e*_1_). The correct rate reflected by *c**(*h*_0_→*e*_0_) is specificity or true negative rate *P*(*h*_0_|*e*_0_). 

Consider the converse channel confirmation measure *b**(*h*_1_→*e*_1_). Now the source is *P*(*e*) instead of *P*(*h*). We may swap *e*_1_ with *h*_1_ in *b**(*e*_1_→*h*_1_) or swap *a* with *d* and *b* with *c* in *f*(*a, b, c, d*) to obtain
(46)$$b^{*}(h_{1}\to e_{1})=\frac{P(h_{1}|e_{1})-P(h_{1}|e_{0})}{P(h_{1}|e_{1})\vee P(h_{1}|e_{0})}=\frac{ad-bc}{a(b+d)\vee b(a+c)}$$
where ˅ is the operator for the maximum of two numbers and is used to replace max( ). There are also four types of converse channel/prediction confirmation formulas with *a*, *b*, *c*, and *d* (see Table 7). Due to Consequent Symmetry, there are the eight types of converse channel/prediction confirmation formulas altogether. 

### 3.4. Eight Confirmation Formulas for Different Antecedents and Consequents

Table 5 shows the positive examples’ and counterexamples’ proportions needed by measures *b** and *c**. 

Table 6 provides four types of confirmation formulas with *a*, *b*, *c*, and *d* for rule *e→h*, where function max( ) is replaced with the operator ˅. 

These confirmation measures are related to the misreporting rates of the rule *e*→*h*. For example, smaller *b**(*e*_1_→*h*_1_) or *c**(*e*_1_→*h*_1_) means that the test shows positive for more uninfected people.

Table 7 includes four types of confirmation measures for *h→e*. 

These confirmation measures are related to the underreporting rates of the rule *h*→*e*. For example, smaller *b**(*h*_1_→*e*_1_) or *c**(*h*_1_→*e*_1_) means that the test shows negative for more infected people. Underreports are more serious problems. 

Each of the eight types of confirmation measures in Table 6 and Table 7 has its consequent-symmetrical form. Therefore, there are 16 types of function *f*(*a*, *b*, *c*, *d*) altogether for confirmation.

In a prediction and converse prediction confirmation formula, the conditions of two conditional probabilities are the same; they are the antecedents of rules so that a confirmation measure *c** only depends on the two numbers of positive examples and counterexamples. Therefore, these measures accord with the Nicod–Fisher criterion.

If we change “˅” into “+” in *f*(*a, b, c, d*), then measure *b** becomes measure *b_F_** = *F*, and measure *c** becomes measure *c_F_**. For example,
*c_F_**(*e*_1_→*h*_1_) = (*a* − *c*)/(*a* + *c*).(47)

### 3.5. Relationship Between Measures b* and F

Measure *b** is like measure *F*. The two measures changes with likelihood ratio *LR*, as shown in Figure 8. 

Measure *F* has four confirmation formulas for different antecedents and consequents [8], which are related to measure *b_F_** as follows:(48)$$F(e_{1}\to h_{1})=\frac{P(e_{1}|h_{1})-P(e_{1}|h_{0})}{P(e_{1}|h_{1})+P(e_{1}|h_{0})}=\frac{ad-bc}{ad+bc+2ac}=b_{F}{}^{*}(e_{1}\to h_{1})$$
(49)$$F(h_{1}\to e_{1})=\frac{P(h_{1}|e_{1})-P(h_{1}|e_{0})}{P(h_{1}|e_{1})+P(h_{1}|e_{0})}=\frac{ad-bc}{ad+bc+2ab}=b_{F}{}^{*}(h_{1}\to e_{1})$$
(50)$$F(e_{0}\to h_{0})=\frac{P(e_{0}|h_{0})-P(e_{0}|h_{1})}{P(e_{0}|h_{0})+P(e_{0}|h_{1})}=\frac{ad-bc}{ad+bc+2bd}=b_{F}{}^{*}(e_{0}\to h_{0})$$
(51)$$F(h_{0}\to e_{0})=\frac{P(h_{0}|e_{0})-P(h_{0}|e_{1})}{P(h_{0}|e_{0})+P(h_{0}|e_{1})}=\frac{ad-bc}{ad+bc+2cd}=b_{F}{}^{*}(h_{0}\to e_{0})$$

*F* is equivalent to *b_F_**. Measure *b** has all the above-mentioned desirable properties as well as measure *F*. The differences are that measure *b** has a greater absolute value than measure *F*; measure *b** can be used for probability predictions more conveniently (see Equation (35)).

### 3.6. Relationships between Prediction Confirmation Measures and Some Medical Test’s Indexes 

Channel confirmation measures are related to likelihood ratios, whereas Prediction Confirmation Measures (PCMs) including converse PCMs are related to correct rates and false rates in the medical test. 

To help us understand the significances of PCMs in the medical test, Table 8 shows that each PCM is related to which correct rate and which false rate.

The false rates related to PCMs are the misreporting rates of the rule *e→h*, whereas the false rates related to converse PCMs are the underreporting rates of the rule *h→e*. For example, False Discovery Rate *P*(*h*_0_|*e*_1_) is also the misreporting rate of rule *e*_1_→*h*_1_; False Negative Rate *P*(*e*_0_|*h*_1_) is also the underreporting rate of rule *h*_1_→*e*_1_.

## 4. Results

### 4.1. Using Three Examples to Compare Various Confirmation Measures

In China’s war against COVID-19, people often ask the question: since the true positive rate, e.g., sensitivity, of NAT is so low (less than 0.5), why do we still believe it? Medical experts explain that though NAT has low sensitivity, it has high specificity, and hence its positive is very believable. 

We use the following two extreme examples (see Figure 9) to explain why a test with very low sensitivity can provide more believable positive than another test with very high sensitivity, and whether popular confirmation measures support this conclusion. 

In Example 1, *b**(*e*_1_→*h*_1_) = (0.1 − 0.01)/0.1 = 0.9, which is very large. In Example 2, *b**(*e*_1_→*h*_1_) = (1 − 0.9)/1 = 0.1, which is very small. The two examples indicate that fewer counterexamples’ existence is more important to *b** than more positive examples’ existence. Measures *F*, *c**, and *c_F_** also possess this characteristic, which is compatible with the Logicality requirement [15]. However, most confirmation measures do not possess this characteristic. 

We supposed *P*(*h*_1_) = 0.2 and *n* = 1000 and then calculated the degrees of confirmation with different confirmation measures for the above two examples, as shown in Table 9, where the base of log for *R* and *L* is 2. Table 9 also includes Example 3 (e.g., Ex. 3), in which *P*(*h*_1_) is 0.01. Example 3 reveals the difference between *Z* and *b** (or *F*). 

Data for Examples 1 and 2 show that *L*, *F* and *b** give Example 1 a much higher rating than Example 2, whereas *M*, *C*, and *N* give Example 2 a higher rating than Example 1 (see red numbers). The excel file for Table 9, Tables 12 and 13 can be find in Appendix A.

In Examples 2 and 3, where *c > a* (counterexamples are more than positive examples), only the values of *c**(*e*_1_→*h*_1_) are negative. The negative values should be reasonable for assessing probability predictions when counterexamples are more than positive examples. 

The data for Example 3 show that when *P*(*h*_0_) = 0.99>>*P*(*h*_1_) = 0.01, measure *Z* is very different from measures *F* and *b** (see blue numbers) because *F* and *b** are independent of *P*(*h*) unlike *Z*.

Although measure *L* (log-likelihood ratio) is compatible with *F* and *b**, its values, such as 3.32 and 0.152, are not intuitionistic as well as the values of *F* or *b**, which are normalizing.

### 4.2. Using Measures b* to Explain Why And How CT is also Used to Test COVID-19

The COVID-19 outbreak in Wuhan of China in 2019 and 2020 has infected many people. In the early stage, only NAT was used to diagnose the infection. Later, many doctors found that NAT often failed to report the viral infection. Because this test has low sensitivity (which may be less than 0.5) and high specificity, we can confirm the infection when NAT is positive, but it is not good for confirming the non-infection when NAT is negative. That means that NAT-negative is not believable. To reduce the underreports of the infection, CT gained more attention because CT had higher sensitivity than NAT.

When both NAT and CT were used in Wuhan, doctors improved the diagnosis, as shown in Figure 10 and Table 11. If we diagnose the infection according to confirmation measure *b**, will the diagnosis be the same as the improved diagnosis? Besides NAT and CT, patients’ symptoms, such as fever and cough, were also used for the diagnosis. To simplify the problem, we assumed that all patients had the same symptoms so that we could diagnose only according to the results of NAT and CT.

Reference [34] introduces the sensitivity and specificity of CT that the authors achieved. According to [33,34] and other reports on the internet, the author of this paper estimated the sensitivities and specificities, as shown in Table 10.

Figure 10 was drawn according to Table 10. Figure 10 also shows sensitivities and specificities. For example, the half of the red circle on the right side indicates that the sensitivity of NAT is 0.5.

We use *c*(NAT+) to denote the degree of confirmation of NAT-positive with any measure *c*, and used *c*(NAT−), *c*(CT+), and *c*(CT−) in like manner. Then we have
*b**(NAT+) = [*P*(*e*_1_|*h*_1_) − *P*(*e*_1_|*h*_0_)]/*P*(*e*_1_|*h*_1_) = [0.5 − (1 − 0.95)]/0.5 = 0.9;
*b**(NAT−) = [*P*(*e*_0_|*h*_0_) − *P*(*e*_0_|*h*_1_)]/*P*(*e*_0_|*h*_0_) = [0.95 − (1 − 0.5)]/0.95 = 0.47.

We can also obtain *b**(CT+) = 0.69 and *b**(CT−) = 0.73 in like manner (see Table 11). 

If we only use the positive or negative of NAT as the final positive or negative, we confirm the non-infection as NAT shows negative. According to measure *b**, if we use both results of NAT and CT, when NAT shows a negative whereas CT shows positive, the final diagnosis should be positive (see blue words in Table 11) because *b**(CT+) = 0.69 is higher than *b**(NAT−) = 0.47. This diagnosis is the same as the improved diagnosis in Wuhan.

Assuming the prior probability of the infection *P*(*e*_1_) = 0.25, the author calculated the various degrees of confirmation with different confirmation measures for the same sensitivities and specificities, as shown in Table 12.

If there is a “No” under a measure, this measure will result in a different diagnosis from the improved diagnosis. The red numbers mean that *c*(CT+) < *c*(NAT−) or *c*(NAT+)<*c*(CT−). Measures *D, M*, and *F*, as well as *b**, are consistent with the improved diagnosis. If we change *P*(*h*_1_) from 0.1 to 0.6, we will find that measure *M* is also not consistent with the improved diagnosis. If we believe a test-positive or test-negative when its degree of confirmation is greater than 0.2, then *D* is also undesirable, and only measures *F* and *b** satisfy our requirements.

The above sensitivities and specificities in Table 10 were not specially selected. When NAT-sensitivity changed between 0.3 and 0.7, or CT-sensitivity changed between 0.6 and 0.9, it was the same that only measures *D, F*, and *b** were consistent with the improved diagnosis. 

Measure *c** is also not suitable for the diagnosis because it reflects correctness and cannot reduce the underreports of the infection. Yet, the underreports of the infection will cause greater loss than the misreports of the infection.

### 4.3. How Various Confirmation Measures are Affected by Increments Δa and Δd 

The following example is used to check if we can use popular confirmation measures to explain that a black raven can confirm “Ravens are black” more strongly than a piece of white chalk.

Table 13 shows the degrees of confirmation calculated with nine different measures. First, we supposed *a* = *d* = 20 and *b* = *c* = 10 to calculate the nine degrees of confirmation. Next, we only replaced *a* with *a* + 1 to calculate the nine degrees. Last, we only replaced *d* with *d* + 1 to calculate them. 

The results must have exceeded many researchers’ expectations. Table 13 indicates that all measures except *c** (see blue numbers) cannot ensure that Δ*a* = 1 increases *f*(*a, b, c, d*) more than Δ*d* = 1. If we change *b* and *c* between 1 and 19, all measures except *c**, *S*, and *N* cannot ensure Δ*f*/Δ*a≥*Δ*f*/Δ*d*. When *b>c*, measures *S* and *N* also cannot ensure Δ*f*/Δ*a≥*Δ*f*/Δ*d*. The cause for measures *D* and *M* is that Δ*d* = 1 decreases *P*(*h*_1_) and *P*(*e*_1_) more than increasing *P*((*h*_1_|*e*_1_) and *P*(*e*_1_|*h*_1_). The causes for other measures except *c** are similar.

## 5. Discussions

### 5.1. To Clarify the Raven Paradox

To clarify the Raven Paradox, some researchers including Hemple [3] affirm the Equivalence Condition and deny the Nicod–Fisher criterion; some researchers, such as Scheffler and Goodman [35], affirm the Nicod–Fisher criterion and deny the Equivalence Condition. There are also some researchers who do not fully affirm the Equivalence Condition or the Nicod–Fisher criterion. 

First, we consider measure *F* to see if we can use it to eliminate the Raven Paradox. The difference between *F*(*e*_1_→*h*_1_) and *F*(*h*_0_→*e*_0_) is that their counterexamples are the same, yet, their positive examples are different. When *d* increases to *d*+Δ*d*, *F*(*e*_1_→*h*_1_) and *F*(*h*_0_→*e*_0_) unequally increase. Therefore,
though measure *F* denies the Equivalence Condition, it still affirms that Δ*d* affects both *F*(*e*_1_→*h*_1_) and *F*(*h*_0_→*e*_0_);measure *F* does not accord the Nicod–Fisher criterion.

Measure *b** is like *F*. The conclusion is that measures *F* and *b** cannot eliminate our confusion about the Raven Paradox. 

After inspecting many different confirmation measures from the perspective of the rough set theory, Greco et al. [15] conclude that Nicod criterion (e.g., the Nicod–Fisher criterion) is right, but it is difficult to find a suitable measure that accords with the Nicod criterion. However, many researchers still think that the Nicod criterion is incorrect; it accords with our intuition only because a confirmation measure *c*(*e*_1_→*h*_1_) can evidently increase with *a* and slightly increase with *d*. After comparing different confirmation measures, Fitelson and Hawthorne [28] believe that the likelihood ratio may be used to explain that a black raven can confirm “Ravens are black” more strongly than a non-black non-raven thing. 

Unfortunately, Table 13 shows that the increments of all measures except *c** caused by Δ*d* = 1 are greater than or equal to those caused by Δ*a* = 1. That means that these measures support the conclusion that a piece of white chalk can confirm “Ravens are black” more strongly than (or as well as) a black raven. Therefore, these measures cannot be used to clarify the Raven Paradox. 

However, measure *c** is different. Since *c**(*e*_1_→*h*_1_) = (*a − c*)/(*a*˅*c*) and *c**(*h*_0_→*e*_0_) = (*d* − *c*)/(*d*˅*c*), the Equivalence Condition does not hold, and measure *c** accords with the Nicod–Fisher criterion very well. Hence, the Raven Paradox does not exist anymore according to measure *c**.

### 5.2. About Incremental Confirmation and Absolute Confirmation

In Table 13, if the initial numbers are *a* = *d* = 200 and *b* = *c* = 100, the increments of all measures caused by Δ*a* = 1 will be much less than those in Table 13. For example, *D*(*e*_1_→*h*_1_) increases from 0.1667 to 0.1669; *c**(*e*_1_→*h*_1_) increase from 0.5 to 0.5025. The increments are about 1/10 of those in Table 13. Therefore, the increment of the degree of confirmation brought about by a new example is closely related to the number of old examples or our prior knowledge.

The absolute confirmation requires that
the sample size *n* is big enough;each example is selected independently;examples are representative.

Otherwise, the degree of confirmation calculated is unreliable. We need to replace the degree of confirmation with the degree interval of confirmation, such as [0.5, 1] instead of 1.

### 5.3. Is Hypothesis Symmetry or Consequent Symmetry desirable?

Elles and Fitelson defined HS by *c*(*e*, *h*) = −*c*(*e*, −*h*). Actually, it means *c*(*x*, *y*) = −*c*(*x*, −*y*) for any *x* and *y*. Similarly, ES is Antecedent Symmetry, which means *c*(*x*, *y*) = −*c*(−*x*, *y*) for any *x* and *y*. Since *e* and *h* are not the antecedent and the consequent of a major premise from their point of view, they cannot say Antecedent Symmetry and Consequent Symmetry. Consider that *c*(*e*, *h*) becomes *c*(*h*, *e*). According the literal meaning of HS (Hypothesis Symmetry), one may misunderstand HS as shown in Table 14.

For example, the misunderstanding happens in [8,19], where the authors call *c*(*h, e*) = −*c*(*h*, −*e*) ES. However, it is in fact HS or Consequent Symmetry. In [19], the authors think that *F*(*H*, *E*) (where the right side is evidence) should have HS: *F*(*H*, *E*) = −*F*(−*H*, *E*), whereas *F*(*E*, *H*) should have ES: *F*(*E*, *H*)= −*F*(−*E*, *H*). However, this “ES” does not accord with the original meaning of ES in [14]. Both *F*(*H*, *E*) and *F*(*E*, *H*) possess HS instead of ES. The more serious thing because of the misunderstanding is that [19] concludes that ES and EHS (e.g., *c*(*H*, *E*) = *c*(−*H*, −*E*)), as well as HS, are desirable, and hence, measures *S*, *N*, and *C* are particularly valuable. 

The author of this paper approves the conclusion of Elles and Fitelson that only HS (e.g., Consequent Symmetry) is desirable. Therefore, it is necessary to make clear that *e* and *h* in *c*(*e*, *h*) are the antecedent and the consequent of the rule *e*→*h*. To avoid the misunderstanding, we had better replace *c*(*e*, *h*) with *c*(*e*→*h*) and use “Antecedent Symmetry” and “Consequent Symmetry” instead of “Evidence Symmetry” and “Hypothesis Symmetry”. 

### 5.4. About Bayesian Confirmation and Likelihoodist Confirmation 

Measure *D* proposed by Carnap is often referred to as the standard Bayesian confirmation measure. The above analyses, however, show that *D* is only suitable as a measure for selecting hypotheses instead of a measure for confirming major premises. Carnap opened the direction of Bayesian confirmation, but his explanation about *D* easily lets us confuse a major premise’s evidence (a sample) and a consequent’s evidence (a minor premise).

Greco et al. [19] call confirmation measures with conditional probability *p*(*h|e*) as Bayesian confirmation measures, those with *P*(*e|h*) as Likelihoodist confirmation measures, and those for *h*→*e* as converse Bayesian/Likelihoodist confirmation measures. This division is very enlightening. However, the division of confirmation measures in this paper does not depend on symbols, but on methods. The optimized proportion of the believable part in the truth function is the channel confirmation measure *b**, which is similar to the likelihood ratio, reflecting how good the channel is. The optimized proportion of the believable part in the likelihood function is the prediction confirmation measure *c**, which is similar to the correct rate, reflecting how good the probability prediction is. The *b** may be called the logical Bayesian confirmation measure because it is derived with Logical Bayesian Inference [17], although *P*(*e|h*) may be used for *b**. The *c** may be regarded as the likelihoodist confirmation measure, although *P*(*h|e*) may be used for *c**. 

This paper also provides converse channel/prediction confirmation measures for rule *h*→*e*. Confirmation measures *b**(*e*→*h*) and *c**(*e*→*h*) are related to misreporting rates, whereas converse confirmation measures *b**(*h*→*e*) and *c**(*h*→*e*) are related to underreporting rates. 

### 5.5. About the Certainty Factor for Probabilistic Expert Systems

The Certainty Factor, which is denoted by *CF*, was proposed by Shortliffe and Buchanan for a backward chaining expert system [7]. It indicates how true an uncertain inference *h*→*e* is. The relationship between measures *CF* and Z is *CF*(*h*→*e*) = *Z*(*e*→*h*) [36].

As pointed out by Heckerman and Shortliffe [36], the Certainty Factor method has been widely adopted in rule-based expert systems, it also has its theoretical and practical limitations. The main reason is that the Certainty Factor method is not compatible with statistical probability theory. They believe that the belief-network representation can overcome many of the limitations of the Certainty Factor model; however, the Certainty Factor model is simpler than the belief-network representation; it is possible to combine both to develop simpler probabilistic expert systems. 

Measure *b**(*e*_1_→*h*_1_) is related to the believable part of the truth function of predicate *e*_1_(*h*). It is similar to *CF*(*h*_1_→*e*_1_). The differences are that *b**(*e*_1_→*h*_1_) is independent of *P*(*h*) whereas *CF*(*h*_1_→*e*_1_) is related to *P*(*h*); *b**(*e*_1_→*h*_1_) is compatible with statistical probability theory whereas *CF*(*h*_1_→*e*_1_) is not.

Is it possible to use measure *b** or *c** as the Certainty Factor to simplify belief-networks or probabilistic expert systems? This issue is worth exploring.

### 5.6. How Confirmation Measures F, b*, and c* are Compatible with Popper’s Falsification Thought

Popper affirms that a counterexample can falsify a universal hypothesis or a major premise. However, for an uncertain major premise, how do counterexamples affect its degree of confirmation? Confirmation measures *F, b**, and *c** can reflect the importance of counterexamples. In Example 1 of Table 9, the proportion of positive examples is small, and the proportion of counterexamples is smaller still, so that the degree of confirmation is large. This example shows that to improve the degree of confirmation, it is not necessary to increase the conditional probability *P*(*e*_1_|*h*_1_) (for *b**) or *P*(*h*_1_|*e*_1_) (for *c**). In Example 2 of Table 9, although the proportion of positive examples is large, the proportion of counterexamples is not small so that the degree of confirmation is very small. This example shows that to raise degree of confirmation, it is not sufficient to increase the posterior probability. It is necessary and sufficient to decrease the relative proportion of counterexamples. 

Popper affirms that a counterexample can falsify a universal hypothesis, which can be explained by that for the falsification of a strict universal hypothesis, it is important to have no counterexample. Now for the confirmation of a universal hypothesis that is not strict or uncertain, we can explain that it is important to have fewer counterexamples. Therefore, confirmation measures *F*, *b**, and *c** are compatible with Popper’s falsification thought. 

Scheffler and Goodman [35] proposed selective confirmation based on Popper’s falsification thought. They believe that black ravens support "Ravens are black" because black ravens undermine "Ravens are not black". Their reason why non-black ravens support "Ravens are not black" is that non-black ravens undermine the opposite hypothesis "Ravens are black". Their explanation is very meaningful. However, they did not provide the corresponding confirmation measure. Measure *c**(*e*_1_→*h*_1_) is what they need.

## 6. Conclusions

Using the semantic information and statistical learning methods and taking the medical test as an example, this paper has derived two confirmation measures *b**(*e →h*) and *c**(*e →h*). The measure *b** is similar to the measure *F* proposed by Kemeny and Oppenheim; it can reflect the channel characteristics of the medical test like the likelihood ratio, indicating how good a testing means is. Measure *c**(*e→h*) is similar to the correct rate but varies between −1 and 1. Both *b** and *c** can be used for probability predictions. The *b** is suitable for predicting the probability of disease when the prior probability of disease is changed. Measures *b** and *c** possess symmetry/asymmetry proposed by Elles and Fitelson [14], monotonicity proposed by Greco et al. [16], normalizing property (between −1 and 1) suggested by many researchers. The new confirmation measures support absolute confirmation instead of incremental confirmation. 

This paper has shown that most popular confirmation measures cannot help us diagnose the infection of COVID-19, but measures *F* and *b** and the like, which are the functions of likelihood ratio, can. It has also proved that popular confirmation measures did not support the conclusion that a black raven could confirm more strongly than a non-black non-raven thing, such as a piece of chalk. It has shown that measure *c** could definitely deny the Equivalence Condition and exactly reflect Nicod–Fisher Criterion, and hence, could be used to eliminate the Raven Paradox. The new confirmation measures *b** and *c** as well as *F* indicates that fewer counterexamples’ existence is more important than more positive examples’ existence; therefore, measures *F*, *b**, and *c** are compatible with Popper’s falsification thought. 

When the sample is small, the degree of confirmation calculated by any confirmation measure is not reliable, and hence, the degree of confirmation should be replaced with the degree interval of confirmation. We need further studies combining the theory of hypothesis testing. It is also worth conducting further studies ensuring that the new confirmation measures are used as the Certainty Factors for belief-networks. 

## Figures and Tables

**Figure 1 entropy-22-00384-f001:**
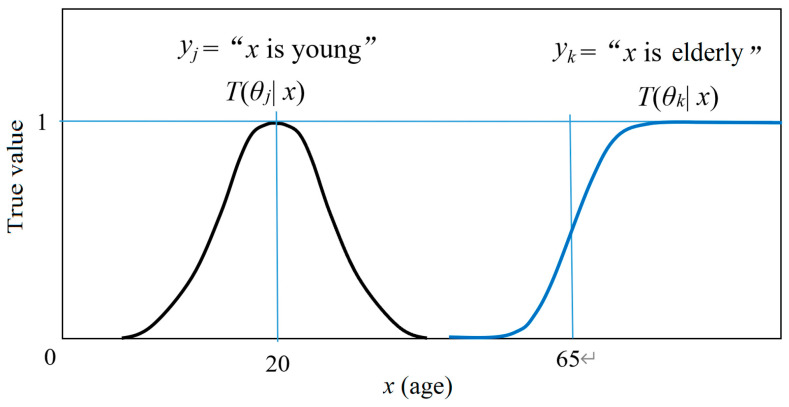
The truth functions of two hypotheses about ages.

**Figure 2 entropy-22-00384-f002:**
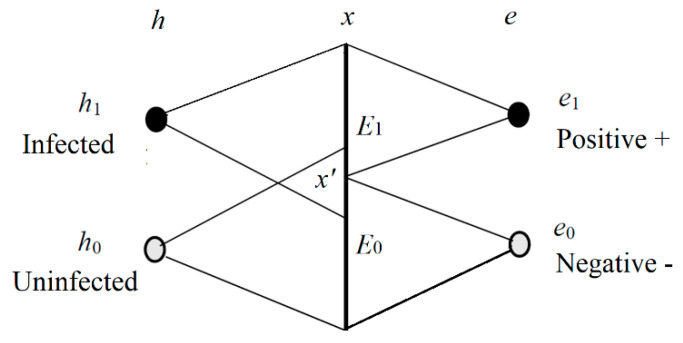
The relationship between Positive/Negative and Infected/Uninfected in the medical test.

**Figure 3 entropy-22-00384-f003:**
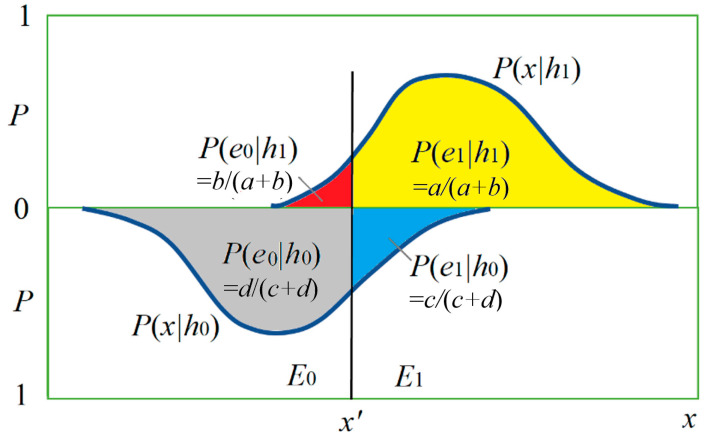
The relationship between two feature distributions and four conditional probabilities for the Shannon channel of the medical test.

**Figure 4 entropy-22-00384-f004:**
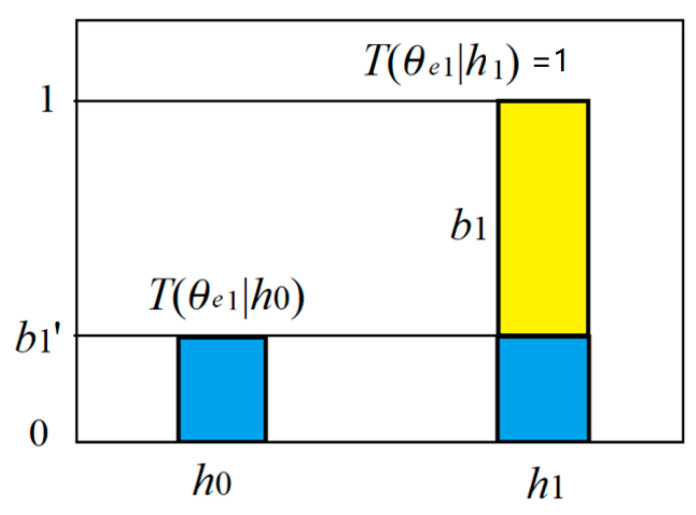
Truth function *T*(*θ_e_*_1_*|h*) includes the believable part with proportion *b*_1_ and the unbelievable part with proportion *b*_1_’ (*b*_1_’ = 1 − |*b*_1_|).

**Figure 5 entropy-22-00384-f005:**
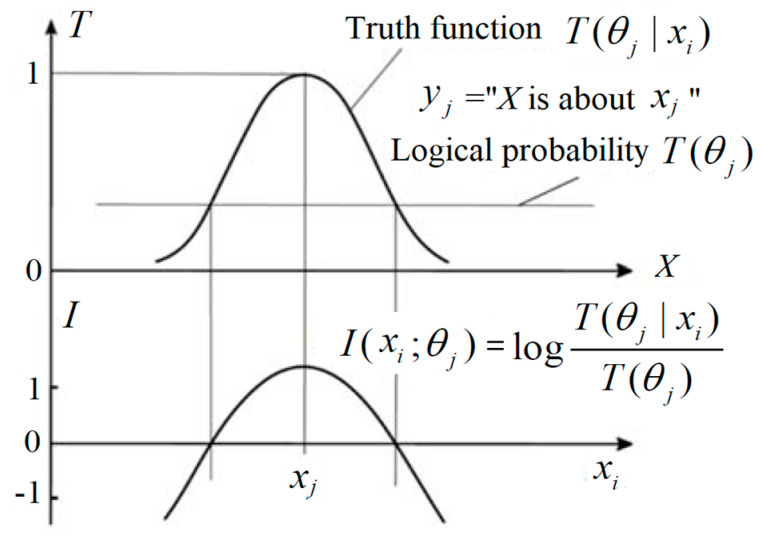
The semantic information conveyed by *y_j_* about *x_i_.*

**Figure 6 entropy-22-00384-f006:**
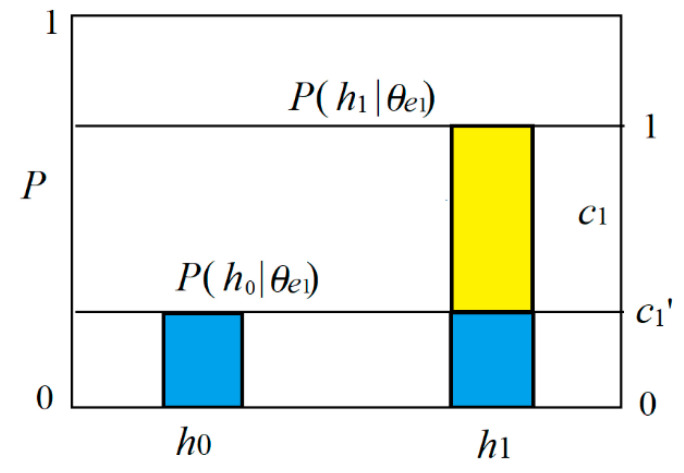
Likelihood function *P*(*h*|*θ_e_*_1_) may be regarded as a believable part plus an unbelievable part.

**Figure 7 entropy-22-00384-f007:**
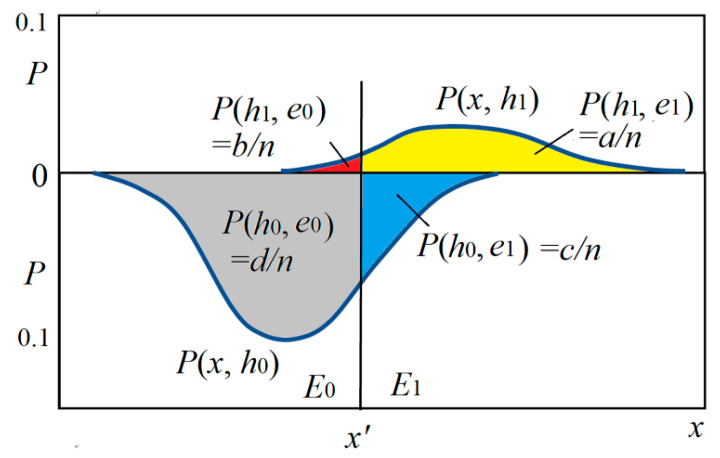
The numbers of positive examples and counterexamples for *c**(*e*_0_→*h*_0_) (see the left side) and *c**(*e*_1_→*h*_1_) (see the right side).

**Figure 8 entropy-22-00384-f008:**
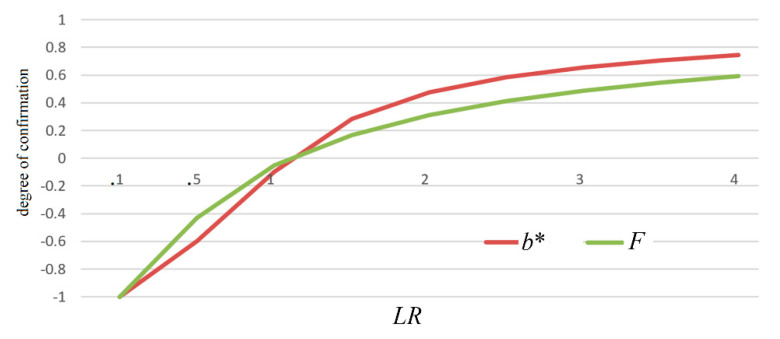
Measures *b** and *F* change with likelihood ratio *LR*.

**Figure 9 entropy-22-00384-f009:**
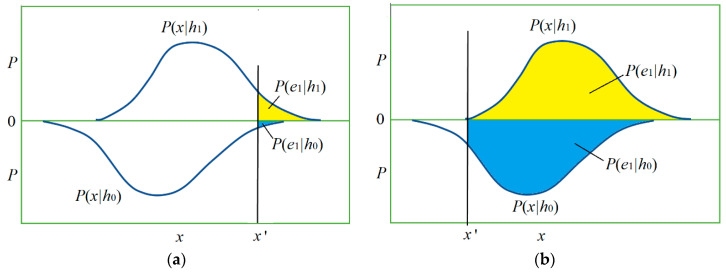
How the proportions of positive examples and counterexamples affect *b**(*e*_1_→*h*_1_). (**a**) Example 1: positive examples’ proportion is *P*(*e*_1_,|*h*_1_) = 0.1, and counterexamples’ proportion is *P*(*e*_1_|*h*_0_) = 0.01. **(b)** Example 2: positive examples’ proportion is *P*(*e*_1_,|*h*_1_) = 1, and counterexamples’ proportion is *P*(*e*_1_|*h*_0_) = 0.9.

**Figure 10 entropy-22-00384-f010:**
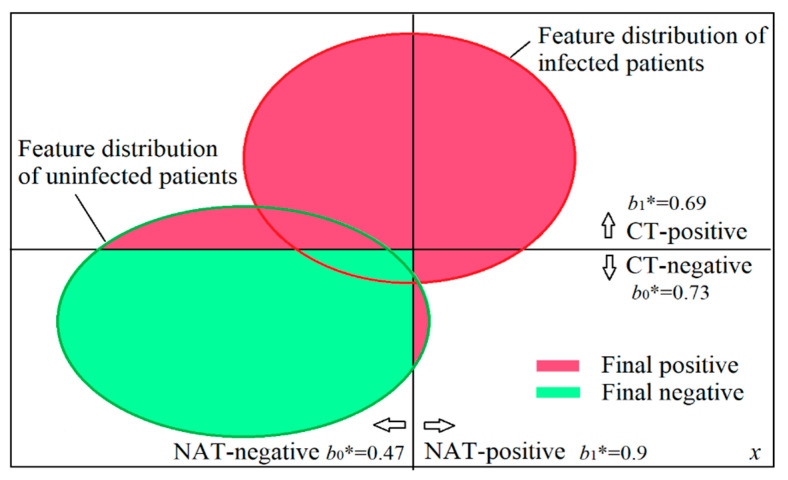
Using both NAT and CT to diagnose the infection of COVID-19 with the help of confirmation measure *b**.

**Table 1 entropy-22-00384-t001:** The numbers of four types of examples for confirmation measures.

	*e*_0_	*e*_1_
*h*_1_	*b*	*a*
*h*_0_	*d*	*c*

**Table 2 entropy-22-00384-t002:** Sensitivit*y* and specificit*y* ascertain a Shannon’s Channel *P*(*e|h*).

	Negative *e*_0_	Positive *e*_1_
Infected *h*_1_	*P*(*e*_0_|*h*_1_) = 1 − sensitivity	*P*(*e*_1_|*h*_1_) = sensitivity
Uninfected *h*_0_	*P*(*e*_0_|*h*_0_) = specificity	*P*(*e*_1_|*h*_0_) = 1 − specificity

**Table 3 entropy-22-00384-t003:** The semantic channel ascertained by *b*_1_’ and *b*_0_’ for the medical test.

	*e*_0_ (Negative)	*e*_1_ (Positive)
*h*_1_ (infected)	*T*(*θe*_0_|*h*_1_) = *b*_0_’	*T*(*θ_e_*_1_|*h*_1_) = 1
*h*_0_ (uninfected)	*T*(*θ_e_*_0_|*h*_0_) = 1	*T*(*θ_e_*_1_|*h*_0_) = *b*_1_’

**Table 4 entropy-22-00384-t004:** Predictive probability *P*(*h*_1_|*θ_e_*_1_) changes with prior probability *P*(*h*_1_) as *b*_1_* = 0.9.

	Common People	Risky Group	High-Risky Group
*P*(*h*_1_)	0.001	0.1	0.25
*P*(*h*_1_|*θ_e_*_1_)	0.002	0.19	0.77

**Table 5 entropy-22-00384-t005:** Eight proportions for calculating *b**(*e→h*) and *c**(*e→h*).

	*e*_0_ (Negative)	*e*_1_ (Positive)
*h*_1_ (infected)	*P*(*e*_0_|*h*_1_) = *b*/(*a* + *b*)	*P*(*e*_1_|*h*_1_) = *a*/(*a* + *b*)
*h***_0_ (uninfected)	*P*(*e*_0_|*h*_0_) = *d*/(*c* + *d*)	*P*(*e*_1_|*h*_0_) = *c*/(*c* + *d*)
*h*_1_ (infected)	*P*(*h*_1_|*e*_0_) = *b*/(*b* + *d*)	*P*(*h*_1_|*e*_1_) = *a*/(*a* + *c*)
*h***_0_ (uninfected)	*P*(*h*_0_|*e*_0_) = *d*/(*b* + *d*)	*P*(*h*_0_|*e*_1_) = *c*/(*a* + *c*)

**Table 6 entropy-22-00384-t006:** Channel/prediction confirmation measures expressed by *a*, *b*, *c*, and *d*.

	*b**(*e→h*) (for Channels, Refer to Figure 3)	*c**(*e→h*) (for Predictions, Refer to Figure 7)
*e*_1_→*h*_1_	$\frac{P(e_{1}|h_{1})-P(e_{1}|h_{0})}{P(e_{1}|h_{1})\vee P(e_{1}|h_{0})}=\frac{ad-bc}{a(c+d)\vee c(a+b)}$	$\frac{P(h_{1}|e_{1})-P(h_{0}|e_{1})}{P(h_{1}|e_{1})\vee P(h_{0}|e_{1})}=\frac{a-c}{a\vee c}$
*e*_0_→*h*_0_	$\frac{P(e_{0}|h_{0})-P(e_{0}|h_{1})}{P(e_{0}|h_{0})\vee P(e_{0}|h_{1})}=\frac{ad-bc}{d(a+b)\vee b(c+d)}$	$\frac{P(h_{0}|e_{0})-P(h_{1}|e_{0})}{P(h_{0}|e_{0})\vee P(h_{1}|e_{0})}=\frac{d-b}{d\vee b}$

**Table 7 entropy-22-00384-t007:** Converse channel/prediction confirmation measures expressed by *a*, *b*, *c*, and *d*.

	*b**(*h*→*e*) (for Converse Channels)	*c**(*h*→*e*) (for Converse Predictions, Refer to Figure 7)
*h*_1_→*e*_1_	$\frac{P(h_{1}|e_{1})-P(h_{1}|e_{0})}{P(h_{1}|e_{1})\vee P(h_{1}|e_{0})}=\frac{ad-bc}{a(b+d)\vee b(a+c)}$	$\frac{P(e_{1}|h_{1})-P(e_{0}|h_{1})}{P(e_{1}|h_{1})\vee P(e_{0}|h_{1})}=\frac{a-b}{a\vee b}$
*h*_0_→*e*_0_	$\frac{P(h_{0}|e_{0})-P(h_{0}|e_{1})}{P(h_{0}|e_{0})\vee P(h_{0}|e_{1})}=\frac{ad-bc}{d(a+c)\vee c(b+d)}$	$\frac{P(e_{0}|h_{0})-P(e_{1}|h_{0})}{P(e_{0}|h_{0})\vee P(e_{1}|h_{0})}=\frac{d-c}{d\vee c}$

**Table 8 entropy-22-00384-t008:** PCMs (Prediction Confirmation Measures) are related to different correct rates and false rates in the medical test [18].

PCM	Correct Rate Positively Related to *c**	False Rate Negatively Related to *c**
*c**(*e*_1_→*h*_1_)	*P*(*h*_1_|*e*_1_): PPV (Positive Predictive Value)	*P*(*h*_0_|*e*_1_): FDR (False Discovery Rate)
*c**(*e*_0_→*h*_0_)	*P*(*h*_0_|*e*_0_): NPV (Negative Predictive Value)	*P*(*h*_1_|*e*_0_): FOR (False Omission Rate)
*c**(*h*_1_→*e*_1_)	*P*(*e*_1_|*h*_1_): Sensitivity or TPR (True Positive Rate)	*P*(*e*_0_|*h*_1_): FNR (False Negative Rate)
*c**(*h*_0_→*e*_0_)	*P*(*e*_0_|*h*_0_): Specificity or TNR (True Negative Rate)	*P*(*e*_1_|*h*_0_): FPR (False Positive Rate)

**Table 9 entropy-22-00384-t009:** Three examples to show the differences between different confirmation measures.

Ex.	*a*, *b*, *c*, *d*	*D*	*M*	*R*	*C*	*Z*	*S*	*N*	*L*	*F*	*b**	*c**
1	20, 180, 8, 792	0.514	0.072	1.84	0.014	0.643	0.529	0.09	**3.32**	**0.818**	**0.9**	0.8
2	200, 0, 720, 80	0.017	0.08	0.12	0.016	0.022	0.217	0.1	**0.152**	**0.053**	**0.1**	−0.722
3	10, 0, 90, 900	0.09	0.9	3.32	0.009	**0.091**	0.1	0.091	3.46	**0.833**	**0.91**	−0.9

**Table 10 entropy-22-00384-t010:** Sensitivities and specificities of NAT (Nucleic Acid Test) and CT for COVID-19.

	Sensitivity	Specificity
NAT	0.5	0.95
CT	0.8	0.75

**Table 11 entropy-22-00384-t011:** Improved diagnosis (for final positive or negative) according to NAT and CT.

	NAT-Negative, *b*_0_* = 0.47	NAT-Positive, *b*_1_* = 0.9
CT-positive, *b*_1_* = 0.69	Final positive (changed)	Final positive
CT-negative, *b*_0_* = 0.73	Final negative	Final positive

**Table 12 entropy-22-00384-t012:** Various confirmation measures for assessing the results of NAT and CT.

	*D*	*M*	*Z*	*S*	*C*	*N*	*F*	*b**	*c**
*c*(NAT-)	0.10	0.11	0.40	0.62	0.08	0.45	0.31	0.47	0.83
*c*(NAT+)	0.52	0.34	0.69	0.62	0.08	0.45	0.82	0.90	0.70
*c*(CT−)	0.17	0.14	0.67	0.43	0.10	0.55	0.58	0.73	0.91
*c*(CT+)	0.27	0.41	0.36	0.43	0.10	0.55	0.52	0.69	0.06
*c*(CT+) > *c*(NAT−)			No	No					No
*c*(NAT+) > *c*(CT−)					No	No			No

**Table 13 entropy-22-00384-t013:** How confirmation measures are affected by Δ*a* = 1 and Δ*d* = 1.

	*f*(*a*, *b*, *c*, *d*)	*a* = *d* = 20 *b* = *c* = 10	Δ*a* = 1 Δ*d* = 0	Δ*d* = 1 Δ*a* = 0	Δ*f*/Δ*a-*Δ*f*/Δ*d*
*D*(*e*_1_→*h*_1_)	*a/*(*a* + *c*) − (*a* + *b*)*/n*	0.167	0.169	0.175	−0.006
*M*(*e*_1_→*h*_1_)	*a/*(*a* + *b*) − (*a* + *c*)*/n*	0.167	0.169	0.175	−0.006
*C*(*e*_1_→*h*_1_)	*a/n −* (*a* + *c*)(*a* + *b*)*/n*^2^	0.083	0.086	0.086	0
*Z*(*e*_1_→*h*_1_)	*D*(*e*_1_→*h*_1_)/[(*c* + *d*)/*n*]	0.333	0.344	0.344	0
*S*(*e*_1_→*h*_1_)	*a/*(*a* + *c*) − *b*(*b + d*)	0.333	0.334	0.344	0
*N*(*e*_1_→*h*_1_)	*a*/(*a* + *b*) *−* *c*/(*c* + *d*)	0.333	0.334	0.344	0
*F*(*e*_1_→*h*_1_)	(*ad-bc*)*/*(*ad* + *bc* + 2*ac*)	0.333	0.340	0.348	−0.007
*LR^+^*	[*a/*(*a* + *b*)]/[*c/*(*c* + *d*)]	2	2.03	2.07	−0.034
*c**(*e*_1_→*h*_1_)	(*a* *−* *c*)/max(*a*, *c*)	**0.5**	**0.524**	**0.5**	**0.024 > 0**

**Table 14 entropy-22-00384-t014:** Misunderstood HS (Hypothesis Symmetry) and ES (Evidence Symmetry).

	*HS* or Consequent Symmetry	*ES* or Antecedent Symmetry
Misunderstood HS	*c*(*e*, *h*) = −*c*(*e*, −*h*)	*c*(*h*, *e*) = −*c*(−*h*, *e*)
Misunderstood ES	*c*(*h*, *e*) = −*c*(*h*, −*e*)	*c*(*e*, *h*) = −*c*(−*e*, *h*)

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
