# Peer review of "Channels’ Confirmation and Predictions’ Confirmation: From the Medical Test to the Raven Paradox"

_entropy, 2020, doi:10.3390/e22040384_

Round 1
Reviewer 1 Report
Dear editor,
I have read with attention the manuscript entitled ‘Channels’ confirmation and Predictions’ Confirmation: From the Medical Test to the Raven Paradox’ by C. Lu and submitted for publication in Entropy.
After a short summary of the paper’s content I shall give my comments on the paper.
----------------------------------------
The article addresses the problem of confirmation measures and discusses the various criteria one may apply to choose one amongst the large set of existing ones or develop a new one if required. Aside from various properties that have been put forward as desirable for confirmation measures, the Raven Paradox (RP) pertains to the fact that most confirmation measures could let one conclude that the observation of a piece of white chalk is evidence of the proposition “if x is a raven then it is black”. It is a paradox because a piece of white chalk is seemingly irrelevant to the existence of black or non-black ravens. The manuscript sets out to derive a novel confirmation measure labelled as ‘prediction confirmation measure’ which disentangles the different kinds of confirmation measures there can be and addresses RP, among other problems, while carrying many of the desirable properties of confirmation measures.
----------------------------------------
The paper is mostly well written, in clear English, and provides background to appreciate the new developments it proposes.
As far as my understanding is concerned, the proposed derivations are valid and supported by examples.
I am very likely to recommend publication of the manuscript but cannot do so just yet. I would like the author to consider the following comments to improve the manuscript first:
The paper is somewhat technical and some notations are only very briefly defined. This does not facilitate the understanding of its content. For example Theta is just defined as a “fuzzy subset on {h0,h1}” without further elaboration on what this means for readers who may not be so familiar with the implications. In general, most notations pertaining to the semantic channel could benefit from a more clearly exposed set of definitions. Such definitions were provided in Ref[17] by the same author. Now, not all definitions of Ref[17] may be of use for the present paper but I am afraid proper mathematical definitions of symbols and concepts and what they entail cannot be skipped from a paper. At the very least, they could be put in Appendix.
Most subsections of the discussion section seem oddly placed. Sections 4.1-4.5 for example contain mostly definitions and discussions about these definitions. Some of them would have been rather useful to appreciate some terms used many times in the precedent sections. My recommendation here would be to put these discussions at the beginning of the paper where all terms and definitions relevant to the discussion should be introduced for optimal clarity. If not at the beginning of the paper, then I believe the author should make sure that terms are not being defined/discussed after they have been heavily used in the paper. On that note, I believe that the acronyms (e.g. RP for Raven Paradox or LR for Likelihood Ratio) should only be mentioned once to prevent impression of repetition.
On that note, I wonder whether what is called Truth function or Logical Probability in the paper is the same as what is called Membership Function in standard expositions of fuzzy sets. Some clarification from the author would be welcome.
The discussion on Popper relies on a distinction between Universal Hypothesis and Uncertain Universal Hypothesis. There seems to be a great difference between the two: a single counter example is enough to refute the former while counter examples only decrease the confirmation level of the latter. I wonder whether this terminology is original or if it has been discussed elsewhere…for example it was my understanding that a hypothesis cannot be certain by definition.
Finally, I believe that there are some typos in the manuscript (the author is welcome correct me if I am mistaken): Figure 3: I believe that P(Theta_e1 | h_1) should be equal to 1 and not P(Theta_e1 | h_0). I believe the name of the function in the final result of Eq. (7) should not be H. For consistency, could the P(h_1|e_0)=b/n etc… in Figure 5 be added to Figure 2 as well. Line 161 I would change “properties are discussed” to “properties discussed”.

Author Response
This file includes
1)The First letter to Editor and Reviewers for explaining the main revisions.
2)The second letter to Reviewer 1.

Reviewer 2 Report
The author presents and discusses some new confirmation measures. The ideas presented could be interesting. However, the paper is written with a style that creates several problems to the reader. Which is the meaning of sentences such as:
"Measures L, F, and b* are the functions of LR"
or
"..., it is still not easy to use F, b*, or other measures to eliminate the RP [Raven Paradox]?
I would suggest the author to rewrite the paper from the scratch, explaining in a simpler and more transparent the confirmation measures he is introducing. For the readability of the paper, I would add a didactic example that could be considered along all the paper to show how the considered confirmation measure work and which are the differences between them. In addition, English language needs to be carefully reconsidered. In addition, some words have to be controlled ("seccedent"?).
I would also avoid speaking of fuzzy sets, since this is not in the tradition of the literature on confirmation measures and, moreover, does not add anything to the content of the paper.
Author Response

(The authors gave the same response as above.)

Round 2
Reviewer 1 Report
Dear editor,
I have read with attention the revised version of the manuscript entitled ‘Channels’ confirmation and Predictions’ Confirmation: From the Medical Test to the Raven Paradox’ by C. Lu and submitted for publication in Entropy.
I shall give below my comments on the paper.
----------------------------------------
I am thankful to the author for having implemented tremendous changes to the manuscript following my comments and those of the other referee.
The paper has now gained greatly in clarity and objectives. A new and timely supplementary part on testing for SARS Cov 2 has also been added to the paper which illustrates the importance of understanding which confirmation measure does what and why it matters to seek new, more reliable, ones.
For these reasons I am happy to recommend the paper for publication in Entropy provided the minor comments below are addressed before actual acceptance:
- I believe the current section 2.4 should be put before section 2.1. That is because a) the notation difference between logical probability and statistical probability is discussed only in section 2.4 while both are being mentioned in the previous sections and b) because the concepts of logical and statistical probabilities are being introduced through a simple example using the recurring example about “x is elderly”.
- Line 305, there is typo for the word ‘channel’.
- Transition from line 532 to 533 seems to be missing. The sentence does not make sense. Maybe the word ‘decrease’ should be somewhere.
- I am quite happy with the clarifying wording of “universal judgment that is not strict or uncertain” but it is slightly ambiguous. I believe it means that the considered universal judgment is not strict/certain and is therefore uncertain (with some probability prediction for example). This would fit the global narrative of the paper. However, if the author means to talk about universal judgments that are neither strict nor uncertain, then I am not quite sure what is meant by the chosen wording.
Reviewer 2 Report
The paper was revised according to what I requested. I believe that the paper can noe be accepted for pubblication. I have only a very marginal remark: I would not write that Hempel discovered the black raven paradox. Rather I would write that Hempel proposed the black raven paradox.
Author Response
Dear Reviewer 2:
Thank you for your accepting the manuscript and for your earlier suggestions, which makes the paper better. I changed “discovered” into “proposed”. See line 42. I also followed Reviewer 1’s instructions to make some changes. After that, I checked all paper.
Best wishes.
Chenguang Lu